

# Drought impact on productivity: Data informed process-based field-scale modeling of a pre-Alpine grassland region

Carolin Boos[1], Sophie Reinermann[2], Raul Wood[3,4,5,6], Ralf Ludwig[3], Anne Schucknecht[7], David Kraus[1], Ralf Kiese[1]

[1]Institute of Meteorology and Climate Research, Atmospheric Environmental Research (IMK-IFU) Karlsruhe Institute of Technology, Garmisch-Partenkirchen, 82467, Germany
[2]Department of Remote Sensing, Institute of Geography and Geology, University of Wuerzburg, Wuerzburg, 97074, Germany
[3]Department of Geography, Ludwig-Maximilans-Universität München, Munich, 80333, Germany
[4]WSL Institute for Snow and Avalance Research SLF, Flüelastr. 11, 7260 Davos Dorf, Switzerland
[5]Climate Change, Extremes, and Natural Hazards in Alpine Regions Research Centre CERC, 7260 Davos Dorf, Switzerland
[6]Institute for Atmospheric and Climate Science, ETH Zurich, Zurich, Switzerland
[7]OHB System AG, Wessling, 82234, Germany

*Correspondence to*: Carolin Boos (carolin.boos@kit.edu)

**Abstract.** Grasslands are the basis for milk and meat production in alpine and pre-alpine regions, where climate warming is occurring twice as fast as in global average. Warmer and drier conditions have been found to lead to versatile effects on grassland productivity and yields depending on pedo-climatic conditions. Experimentally, it has been discovered that higher and cooler elevations benefit from warming in the absence of drought, whereas lower elevations are more vulnerable to yield losses under climate change. These findings are based on sites covering only a few discrete climatic, soil, and management conditions. This limitation is overcome in the present study, where we compiled a highly detailed field-scale dataset including cutting dates (2018-2020) from remote sensing, informing regional grassland management routines of the biogeochemical model LandscapeDNDC which was applied in the pre-Alpine Ammer region (530 m a.s.l to 2200 m a.s.l, 4600 km²) in southern Germany. The strongest predictor of yields was the management intensity with an average yield increase of 1.2 t ha$^{-1}$ a$^{-1}$ per additional cut and associated manure application. At the regional scale for 3- and 5-cut fields, yields decrease on average with increasing elevation by up to 0.4 t ha$^{-1}$ a$^{-1}$ per 100 m. We found a mean regional yield decrease of 4 % in the drought year 2018 compared to the year 2020 with average climatic conditions. In addition, due to support of mineralization, soil organic carbon had a positive effect on yields, especially in drier years. Yield increases of 0.09 to 0.22 t ha$^{-1}$ a$^{-1}$ per % increase in soil organic carbon were observed. Our results illustrate the complex interactions between management, soil, and climate factors influencing grassland yields, including differences in their importance in drought and non-drought years.



# 1 Introduction

Agriculture in the Alpine and pre-Alpine regions of southern Germany is dominated by permanent grasslands (Kiese et al., 2018; Winkler et al., 2021). Economically, these are used for fodder production and dairy farming with intensities ranging from extensive management with at most three cuts per year to highly intensive management with up to six cuts per year

(Schlingmann et al., 2020; Reinermann et al., 2022). As additional ecosystem services, permanent grasslands support water retention and reduce erosion and especially extensively used grasslands support biodiversity (Wilson et al., 2012; Väre et al., 2003) and (White, 2000; Bengtsson et al., 2019). Pre-Alpine grassland soils contain large amounts of organic carbon, which can be explained by historic organic fertilizer inputs as well as cold and wet climatic conditions reducing mineralization (Wiesmeier et al., 2013). Following the Köppen-Geiger classification, the climate in the northern pre-Alps is characterized as

a temperate oceanic climate (Cfb) (Kottek et al., 2006). Air temperature decreases and precipitation increases with elevation (Kiese et al., 2018). Currently, the Alps and pre-Alps are warming at about twice the pace as the global average (Auer et al., 2007; Kiese et al., 2018). With climate change, summers are expected to get drier with a reduction in precipitation of roughly 30 % comparing 1961-1990 and 2070-2100 (Smiatek et al., 2009). This reduction of summer precipitation is accompanied by an increase in frequency and intensity of drought events (Calanca, 2007; Gobiet et al., 2014), which often coincide with

heat waves (Bevacqua et al., 2022). Therefore, climate change impacting alpine grasslands, can be divided into two categories: i) Long-term continuous trends of rising $CO_2$-concentrations, increasing temperatures, and shifts in precipitation regimes and ii) extreme, rather discrete, events including heavy rain, drought, heat-waves, and compound events. In order to allow adaption to future conditions, it is crucial to understand the associated effects on grassland ecosystem functioning and yields.

Experimentally, climate change is often mimicked by a space for time approach, i.e. the translocation of intact plant-soil cores to lower elevations. If such an experiment is pursued for many years, it rather targets the long-term impact of continuous trends i) on grasslands (Kiese et al., 2018; Schlingmann et al., 2020; Wang et al., 2016, 2021), whereas if the first weeks or season after a translocation to a lower altitude are evaluated, the climatic effect is similar to ii) a discrete drastic change of climatic conditions (Berauer et al., 2019; De Boeck et al., 2016). The latter was done by De Boeck et al., who

moved soil cores from 2440m downwards along a 1700m gradient in the central Alps. No significant effect on above ground biomass was observed from only raised temperatures. However, under additional drought, productivity decreased at locations with higher temperatures. Another short-term translocation experiment found increases in biomass for translocations from Alpine to montane sites, and decreases for translocations to a much drier colline site (Berauer et al., 2019). These findings are supported by a field-based survey, which discovered a positive effect on peak biomass from

increasing mean summer maximum temperatures in the Central French Alps at elevations between 1552 and 2442 m a.s.l. (Grigulis and Lavorel, 2020). Only in extremely hot summers (2015, 2017, 2018) biomass was reduced, which was suggested to be linked to a regime shift, potentially caused by heat stress, drought, physiological damage by high light intensity, or accelerated phenology. The evaluation of climatic effects for different management intensities has been studied



rather sparsely. In a translocation experiment in pre-Alpine Bavaria, Germany, a larger yield decrease has been found for
intensive (-1.5 t C ha$^{-1}$ a$^{-1}$, -27 %) than for extensive (-0.8 t C ha$^{-1}$ a$^{-1}$, -21 %) management in a dry season in absolute as well
as relative numbers (Wang et al., 2021).

Experimental setups are normally limited to a few pedo-climatic conditions as well as management options, and are mostly
conducted at ambient $CO_2$-concentrations. Further, there can be severe methodological problems from soil disturbance
during translocation or boundary effects. These limitations can be overcome, if process-based models are calibrated on such
field measurements and are used for temporal and spatial upscaling within scenario simulations. Under the representative
concentration pathway (RCP) 8.5, Petersen et al. (2021) found yield increases of 15 % for pre-Alpine sites in Southern
Germany by the end of the century using the biogeochemical model LandscapeDNDC (Haas et al., 2013), which the authors
mainly linked to an expansion of the growing seasons. In contrast, for grasslands in central Europe a yield decrease of 24 %
comparing the end of the 21$^{st}$ century and a reference period (1978–2004) was found (Carozzi et al., 2022) via a gridded
approach with the PaSim model (Riedo et al., 1998). These losses were correlated with increasing air temperatures and
decreasing precipitation amounts as present in the RCP 8.5 after 2050. Both studies include effects of increasing atmospheric
$CO_2$-concentrations, which were reported to have a positive impact on yields (Andresen et al., 2018). Another large-scale
approach to analyze climatic effects on grassland yields is the use of remote sensing [RS] techniques. For instance,
Straffelini et al. (2024) found that in the extreme drought and high-temperature summer of 2022, nearly a quarter of
European mountain grasslands were negatively affected.

The variety of effects found for grassland productivity due to warming, changes in precipitation, and in part also rising
atmospheric $CO_2$, shows the wide range of systems present in the Alps and pre-Alps with comparably strong gradients in
pedo-climatic parameters and management regimes varying on small spatial scales. It also points towards the fact that
impacts from climate change on grassland productivity originate not only from the direct effects on plant growth, but also
from indirect effects due to the alteration of soil nitrogen and carbon turnover. For example, Wang et al. (2016) found a
significant increase in nitrogen mineralization rates and thus nitrogen availability in a translocation experiment, which has a
positive effect on plant growth. A global meta-analysis discovered increased carbon fluxes, like a 10 % increase of net
primary productivity at a mean warming of 2 °C, but also raised soil respiration (Wang et al., 2019).

In the present study, our main objective is to assess the impact of the exceptional warm and dry year 2018 on grassland
yields in the Ammer region (530-2200 m a.s.l., 4600 km²) in Southern Germany with the process based biogeochemical
LandscapeDNDC model, to compare the simulation results to the two subsequent climatically rather normal years, and to
link yields to influential environmental parameters. To this end, we compiled a highly detailed soil, climate, and
management dataset representing grassland fields (n=28202) in the study region with a much higher spatial resolution than
previous studies executed on the European scale. This allows analyzing the large- and small-scale variability of conditions
present in the study region which is essential also to inform practitioners who make management decision on field to farm
scale. The high degree of detail also allows to differentiate climate impacts on different management intensities. We
hypothesize that i) yields decrease under drought conditions in 2018 and that this effect increases with the management





intensity, that ii) sites at higher elevation with more rainfall show a smaller reduction in yields under drought than sites at lower elevation with less rainfall, and that iii) the benefit of soil organic matter on water storage and nitrogen supply, both

supporting plant growth, is particularly large under drought conditions.

## 2 Material and methods

### 2.1 Model description

LandscapeDNDC is a process-based biogeochemical model framework, which simulates the carbon, nitrogen, and water fluxes, in hand with plant growth for various terrestrial ecosystems (Haas et al., 2013). In the last ten years,

LandscapeDNDC was applied and validated for croplands (Haas et al., 2022; Kraus et al., 2022; Smerald et al., 2022), forests (Grote et al., 2020; Nadal-Sala et al., 2021, 2024) and grasslands (Houska et al., 2017; Molina-Herrera et al., 2016; Petersen et al., 2021). For the latter, LandscapeDNDC has been used in particular for European sites. Model validation has been done by model-data fusion (Keenan et al., 2011) for a non-alpine site Vollnkirchen in Germany (Houska et al., 2017), further the N-cycling of the model has been assessed with N-isotope analysis for another grassland site, Chamau in

Switzerland (Denk et al., 2019). N-loss mitigation scenarios on sites in the UK and Switzerland (Molina-Herrera et al., 2016), as well as plant uptake of nitrate from ground water and responses to elevated $CO_2$ levels for the site Linden in Germany have been assessed (Liebermann et al., 2018). Most recent, investigations on management and yield behavior under climate change scenarios for the pre-Alpine sites Fendt and Graswang have been performed by Petersen et al. (2021).

For compilation, LandscapeDNDC requires the technical framework Crabmeat (**C**oupling and **R**egion**A**lization of

**B**iogeochemical **M**od**E**l **A**pplica**T**ions) providing model communication and I/O facilities. In this study, we ran LandscapeDNDC (revision: 10786, Crabmeat revision: 8136) with an hourly time step. For every grassland site (field or lysimeter) an individual 1-D simulation including a 2-year spin-up was performed. We used the following sub-model set-up: CanopyECM (Grote et al., 2009) as the microclimate module, WatercycleDNDC (Kiese et al., 2011) as the watercycle module, MeTr[x] (Kraus et al., 2015) as the soil-chemistry module, and PlaMo[x] (Kraus et al., 2016) as the physiology module.

CanopyECM is the sub-model determining the temperature in the canopy and soil layers. Also, the radiation distribution in the canopy, which is relevant for the photosynthetic activity is determined in this microclimate module. The hydrology sub-model WatercycleDNDC simulates the water balance of the system and calculates among others evapotranspiration and soil water dynamics. For the soilwater movement, a tipping-bucket approach is used, and water contents at wilting point and field capacity together with the saturated hydraulic conductivity for every soil layer are most sensitive input parameters. The

MeTr[x] sub-model includes all biogeochemical processes which control soil carbon and nitrogen turnover. The physiology sub-model PlaMo[x] employs the PhotoFarquhar model (Ball et al., 1987; Farquhar et al., 1980) for photosynthesis, and allocates carbon to different plant compartments, roots, stem, leaves, and storage depending on the phenology, which is modeled by the concept of growing degree days. Leaf area, root mass distribution, respiration, senescence from age, drought, or frost, and more are calculated. As an input, the model requires time-specific management events like planting, cutting,



harvesting, and fertilizing, each characterized by specific features. For grassland certain aspects, in particular regarding storage and cutting, are different from crops. In this study, we use the species *perennial grass*, which is aimed at describing the whole variety of plants present at the fields with an average parametrization. For perennial grass, storage is built up strongly at the end of the growing season which is used for regrowth in spring. This storage is also used after cutting events. For more details, see Petersen et al. (2021).

## 2.2 Validation and uncertainty assessment

LandscapeDNDC has been calibrated and validated against the grassland yields from intensively managed control lysimeters in Graswang and Fendt from the TERENO pre-Alpine observatory for 2012-2018 (Petersen et al., 2021). Here, we validate the model against an extended dataset including all years up to 2021, and one additional site in Rottenbuch, as well as extensive management at all sites. Graswang is located in an Alpine valley and has the highest elevation of the three sites,

the smallest mean annual temperature [MAT] (6.8 °C) and the highest mean annual precipitation [MAP] (1337 mm). Averages are taken over the reference time 2012-2021. The soil in Graswang is a Fluvic Calceric Cambisol with a soil organic carbon content [SOC] of 6.4±0.6 % in the topsoil. Rottenbuch is the mid-elevation site at 769 m a.s.l. with a MAT of 8.6 °C and a MAP of 1102 mm. The soil is a Cambic Stagnosol, with 4.0±0.3 % SOC. These soil parameters are similar at the site Fendt with the lowest elevation of 595 m a.s.l. with a MAT of 8.9 °C and a MAP of 983 mm. At each site, till 2018,

lysimeters with intensive and extensive management were operated (3 replicates each). From 2019 on, the previously extensively managed lysimeters in Rottenbuch and Fendt were also treated intensively. For more details, see Kiese et al. (2018) and Petersen et al. (2021). We validate the model for harvested dry weight biomass [DWBM] of individual cutting events. To this end, the on-site soil, climate, and management data has been used as model input.

For independent regional yield evaluation, we employ independent regional biomass data from a field campaign conducted

to obtain in-situ data for the validation of RS products in 2019 and 2020 within the Ammer catchment. For details compare (Schucknecht et al., 2020, 2023) and Appendix A. From this dataset, we included fields categorized as meadows and mowing pastures in the Integrated Administration and Control System [INVEKOS]. Furthermore, we only included samples of standing DWBMs as yield proxy if the measurement was conducted within one week before a cutting event. In total, we used 19 yield estimates from the RS field campaign for the comparison with LandscapeDNDC simulations using regional

model input (Sect. 2.3.2 and Sect. 2.3.3).

## 2.3 Regional model drivers

The study region covers mainly the catchment of the river Ammer in Southern Germany and ranges from south west of Munich to Garmisch-Partenkirchen. It is located in the districts of Weilheim-Schongau, Garmisch-Partenkirchen, Landsberg am Lech, Ostallgäu, Bad Tölz-Wolfratshausen, and Starnberg and covers an elevation range from 534 m a.s.l. to 1353 m

a.s.l. We generated regional input data by superimposing information of climate, soil, and management. Single simulation




domains, i.e. fields, originated from INVEKOS data for the year 2019, where 28202 meadows and mowing pastures with a total area of approximative 615 km². These represent the major grassland types and contribute to 75 % of the total grassland (819 km²) area in the study region.

### 2.3.1 Soil

Regional soil information was provided by the Bavarian Environment Agency (Bayerisches Landesamt für Umwelt [LfU], 2022). The heterogeneity of grassland soils within a distinct soil unit (mean area is 0.9 km$^2$) is represented by a set of reference profiles, which are linked to a probability of occurrence. For every profile, detailed information about the individual soil layers is given, i.e. texture, bulk density [BD], stone fraction, organic carbon and nitrogen contents, saturated hydraulic conductivity, pH-values, as well as the water contents at wilting point, and field capacity. We exclude organic as

well as stony and shallow soils, since these are not represented within the validation sites used for LandscapeDNDC. Detailed criteria for exclusion are soil organic carbon [SOC] larger 10 % at any point in the profile, a BD smaller 0.6 g cm$^{-3}$ in the top soil, and a stone content larger 70 % throughout the whole profile.

| description | SOC [%] | N [%] | BD [g cm$^{-3}$] | pH | #fields |
|---|---|---|---|---|---|
| brown earth-lessivé from moraine of Lech glacier | 3.7 | 0.38 | 1.1 | 6.1 | 11139 |
| brown earth-gley from sandy-loamy cap above gravelly loam | 3.4 | 0.22[*] | 1.3 | 5.4 | 3331 |
| pseudogley-brown earth from gravely, loamy, and clayey cover sediments on top of intercalated boulder clay | 4.0 | 0.26[*] | 1.1 | 6.0 | 2569 |
| humus gley from silty-clayey valley sediments above carbonate subsoil | 8.1 | 0.74 | 0.8 | 7.3 | 1860 |
| pseudogley from silty and clayey caps above till | 8.3 | 0.54[*] | 0.8 | 6.3 | 1656 |

**Table 1: Properties of key soil profiles and number of associated fields. SOC, nitrogen content, bulk density, and pH-value are given for the top soil. The values marked with [*] are estimated within the routines of LandscapeDNDC, since no values were**
**provided with the soil profiles from LfU.**

For simulations, only the profile with the highest probability of occurrence was considered. Next, each simulation domain is linked to the largest spatially overlapping soil profile. In total 98 distinct soil profiles are used. The mean (median) organic carbon content in the top soil is 4.5 %, while that of the most common reference soil profile ('brown earth-lessivé from moraine of Lech glacier': 11139 fields) is 3.7 %. Further information on the most relevant soil profiles is given in Table 1.

### 2.3.2 Management

Cutting dates for each simulated field were extracted from Sentinel-2 data for the years 2018 to 2021. (Reinermann et al., 2022, 2023). The numbers of fields (meadows and mowing pastures with valid soil data) with certain numbers of cuts per year are illustrated in a Sankey diagram in Fig. 1. The height of the boxes illustrates the number of fields with x-many cuts in 2018, 2019, and 2020, from left to right. For example, 1945 fields were cut 5 times in 2018, whereas in 2020 it was only
1141 fields. The fluxes inbetween are colored according to the number of cuts in 2018. The dominant management in the



study area is given by 3 cuts, followed by 4 and 2 cuts. The area-weighted mean numbers of cuts in 2018, 2019, and 2020 are 3.21, 3.10, and 3.07, respectively. It is important to note, that in most cases the management on a single field varies between the years. Only 17 % of analyzed fields have the same number of cuts in all three years. Still, for these fields the timing of the cut i.e. the days of the year [DOYs] vary across years. The number of fields in the study region which are intensively

managed (>4 cuts) is rather small, but increased in the drought year 2018 compared to the other years.

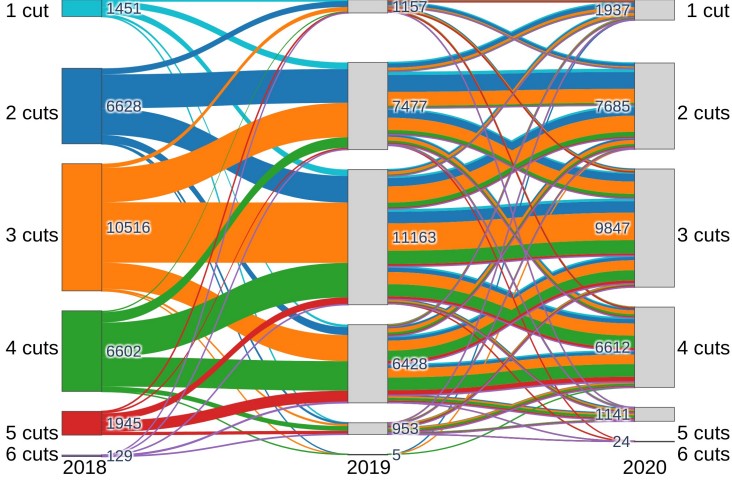

**Figure 1: The number of fields with a certain number of cuts for 2018 to 2020 is illustrated by the height of the boxes and given by the numbers. The fluxes illustrate the variation in management between the years. Colors represent the number of cuts applied in 2018.**

For the spin-up phase of the regional simulations, the RS derived cutting dates from 2020 are used. The frequency and timing of manuring events is implemented in line with the reported local farmer´s grassland management at sites of the TERENO pre-Alpine observatory in 2012-2018 and summarized in Table 2. In the extensive management regime, with less than four cuts in a year, the number of manuring events equals the number of cuts minus 1, whereas for intensive management, the number of manuring events equals the number of cuts. The first fertilization in every year is set to 15

March for all fields with more than 1 cut. For 3-cut fields, the second manure application is applied 3 days after the last cut of the year. For all intensive fields, the manure is applied 3 days after cutting, except for the second cut, where no fertilizer is applied (Petersen et al., 2021). We analyzed the manure events performed on the lysimeters in the TERENO pre-Alpine observatory in 2012–2018, and additionally took expert knowledge into account to derive the amount of slurry that is applied in each simulated manuring event. We performed a linear fit on manure amounts grouped by the total number of manuring

events in a year against the number of manuring events in a year (1st application, 2nd application, etc.) which led to the amounts given in Table 1. No additional mineral fertilizer was applied. On average, in 2018, 119 kg N ha$^{-1}$ were applied, whereas in 2019 it was 112 kg N ha$^{-1}$, and in 2020 111 kg N ha$^{-1}$, due to the variation in the number of cuts between the years. Following from TERENO measurements the applied slurry in the model set-up has a pH-value of 7.6, and a C/N ratio of 8.9 (Petersen et al., 2021).




| #cuts | 1st manure | 2nd manure | 3rd manure | 4th manure | 5th manure | 6th manure |
|---|---|---|---|---|---|---|
| 1 | | | | | | |
| 2 | 15th March 45 | | | | | |
| 3 | 15th March 53 | 3 days after 3rd cut 45 | | | | |
| 4 | 15th March 50 | 3 days after 1st cut 45 | 3 days after 3rd cut 40 | 3 days after 4th cut 36 | | |
| 5 | 15th March 58 | 3 days after 1st cut 53 | 3 days after 3rd cut 49 | 3 days after 4th cut 44 | 3 days after 5th cut 40 | |
| 6 | 15th March 55 | 3 days after 1st cut 53 | 3 days after 3rd cut 51 | 3 days after 4th cut 49 | 3 days after 5th cut 47 | 3 days after 6th cut 45 |

**Table 2: Manure events for different numbers of yearly cuts defined by timing and manure amounts in kg N ha$^{-1}$.**

### 2.3.3 Climate

The climate data was created in the ClimEx project (Climate change and hydrological extreme events-risks and perspectives for water management in Bavaria and Québec; Munich, Germany). We use the developed Sub-Daily Climatological

REFerence data set (SDCLIREF), which is based on hourly and disaggregated daily station measurements from the German weather service, hydrological agencies, and other local authorities. The temporal resolution of the data set is 3 hours. Daily station data was temporally disaggregated by the method of fragments (Poschlod et al., 2018; Westra et al., 2012) to extend the sub-daily record as well as to densify the station network. The in-situ station data were interpolated to a 500 m x 500 m grid by combining a multiple-linear regression considering elevation, exposition, latitude, and longitude, and inverse

distance weighting adapted from (Rauthe et al., 2013). The data includes air temperature, precipitation, relative humidity, global radiation, and wind speed. We aggregated the 3-hourly data into daily values for LandscapeDNDC simulations: For temperature, minimum, maximum, and average values, for precipitation daily sums, and for all other parameters daily means are used. Every field is simulated with climate data from the closest grid cell.

### 2.3.4 Atmospheric CO$_2$ concentration and N deposition

Atmospheric conditions were modeled identically for every field. The atmospheric CO$_2$ concentrations were taken from measurements on top of the Zugspitze provided by the German Environment Agency (Umweltbundesamt, 2024), where monthly mean values are given (403-420 ppm). For wet deposition, constant values were assumed, with 0.9 N–NO$_3$ mg l$^{-1}$ for nitrate and 1.4 N–NH$_4$ mg l$^{-1}$ for ammonium. These values are multiplied by the amount of rainfall at days of precipitation to and nitrogen loads are added to the first soil layer. Yearly N-depositions are comparable to the results from a measurement

campaign in the area (Kirchner et al., 2014).



### 2.4 Aggregation and statistical analysis

For spatial representation and comparison between the years, data is aggregated into hexagons for all fields with the same number of cuts individually. Note that the annual numbers of cuts on individual fields may change between the years. The study region is covered by 381 hexagons of 7.8km². On average, a hexagon includes a total of 37 single fields. For statistical analysis, only hexagons with at least three fields are included. Mean values and according standard deviations [SDs] are weighted by the area of included fields.

We use Python version 3.12.3 in jupyter notebook version 6.0.3 for statistical analysis. The *scipy.stats* package in version 1.13.0 is used to determine Pearson correlation coefficients, confidence intervals, p-values, and linear regressions. For the comparison of mean yields between different years, Welch's t-test with an alternative hypothesis of reduced yields in 2018 and 2019 is chosen, and taken from *statsmodels.stats.weightstats*, in order to weigh by area.

For every field, management, weather data, and harvested biomass vary between the years. We consider all field-year pairs as individual data for the correlation and regression analysis. Surely, they are not fully independent, but the variation in management and weather patterns together with many fields allows for this approximation. We do not include seasonal climate parameters as predictors, because the data from three years is too scarce for a sound analysis covering only a small number of within year patterns. We rather compare the analysis of individual years. We mainly focus on fields with 3 and 5 cuts as representatives of extensive and intensive management.

Overall, the dataset is rather large with up to 81700 field-year data points. For such large datasets, the statistical significance via the p-value becomes rather meaningless, since it always converges to either 0 or 1 if the sample size goes to infinity (Lin et al., 2013). We therefore focus on the effect size and add confidence intervals [CI] for correlation coefficients (Johnson, 1999) to the analysis. The CIs are determined via Fisher's transformation (Fisher, 1915, 1921, 1924) with *scipy.stats.pearsonr* and a confidence level of 0.95.

## 3 Results

### 3.1 Model validation

The temporal development of simulated and measured biomass at cutting events (n=3 replicates) as dry weight for the three sites and two treatments are shown in Fig. 2. The coefficient of determination, calculated for all 238 data points, is $r^2$=0.61 (r=0.78, p<0.0001). The root mean square error [RMSE] is 0.94 t ha$^{-1}$. The regression line between measured and simulated values has a slope of 0.61, hence LandscapeDNDC slightly underestimates low and overestimates high yields at cutting events, i.e. the decrease in biomass with every cut throughout the year is stronger in LandscapeDNDC than represented in the measurements. Under the exclusion of the year 2013, where measurement problems may have occurred (Petersen et al., 2021), statistical measures improve further ($r^2$=0.71, r=0.84, p<0.0001). If the data is separated into extensive and intensive management the values are $r^2$=0.51 (r=0.72, p<0.0001) and $r^2$=0.66 (r=0.81, p<0.0001), respectively. If one compares yearly





sums per management and site, and again exclude the year 2013, $r^2$ is 0.47, the slope is 0.75, and the RMSE is 1.5 t ha$^{-1}$ a$^{-1}$. A direct comparison of measured and simulated yields (for single cuts and annual sum) are shown in Fig. A1 in Appendix B.

275

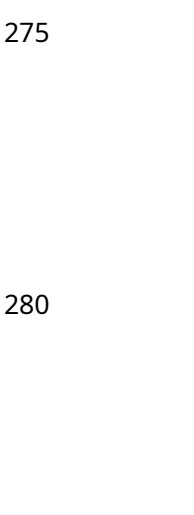

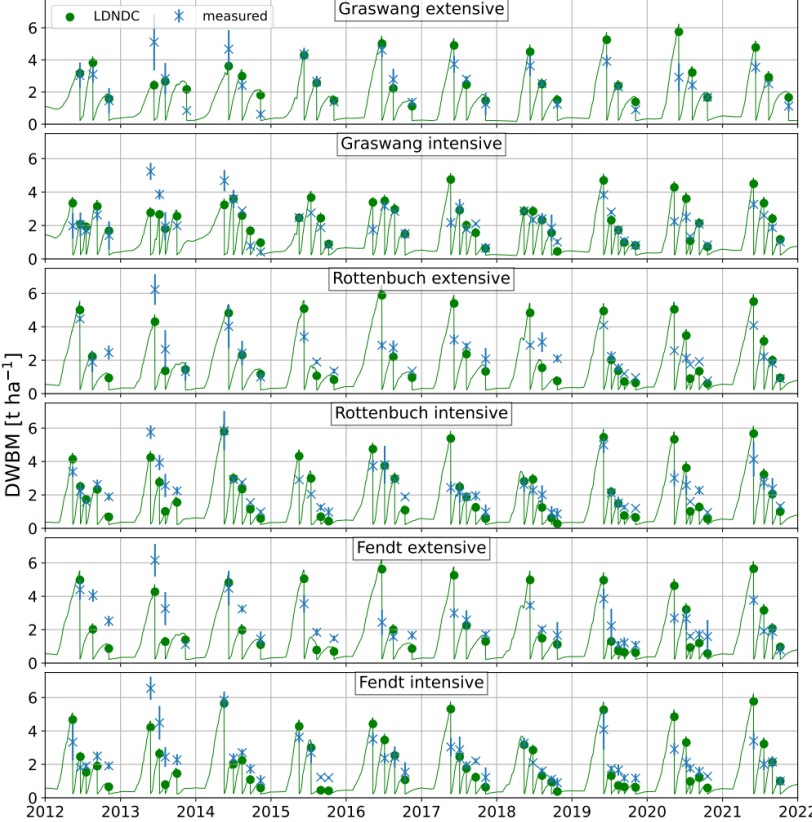

**Figure 2: DWBM simulated with LandscapeDNDC and measured in the TERENO pre-Alpine observatory for the sites Graswang, Rottenbuch, and Fendt for extensive and intensive management, respectively. For the measurements the mean values and SDs from 3 replicates as error bar are shown.**

Over all 10 years (2012-2021), the mean simulated yields are 3 % above the measured yields. For 2018 and 2019, the sum of simulated yields over all treatments and sites per year, is 2 % below the measured amounts. For 2020, the simulated yield sums are 21 % above the measured sums.

## 3.2 Regional uncertainty evaluation

In Fig. 3, the measured DWBM from the independent campaign used for RS validations are plotted against the according values determined by LandscapeDNDC. For the individual measurements, shown on the left (a), the regression gives a slope of 0.26, hence large yields are overestimated by LandscapeDNDC, whereas lower yields are underestimated. The coefficient of determination is $r^2$=0.16 with p=0.09 and the RMSE is 1.37 t ha$^{-1}$. In Fig. 3b, the same data is aggregated into yearly sums per plot. The measures of performance are $r^2$=0.67, p=0.0006, a slope of 0.76, and a RMSE of 1.46 t ha$^{-1}$ a$^{-1}$. Note, that the




relative RMSE is smaller for the yearly aggregated results with 0.31 compared to 0.43 for the individual data. If one considers the total yearly sums, LandscapeDNDC reaches values 3.0 % and 11.1 % below the measured ones in 2019 (11 data points) and 2020 (8 data points), respectively. For all data together, simulation results are 6.9 % below measurements.

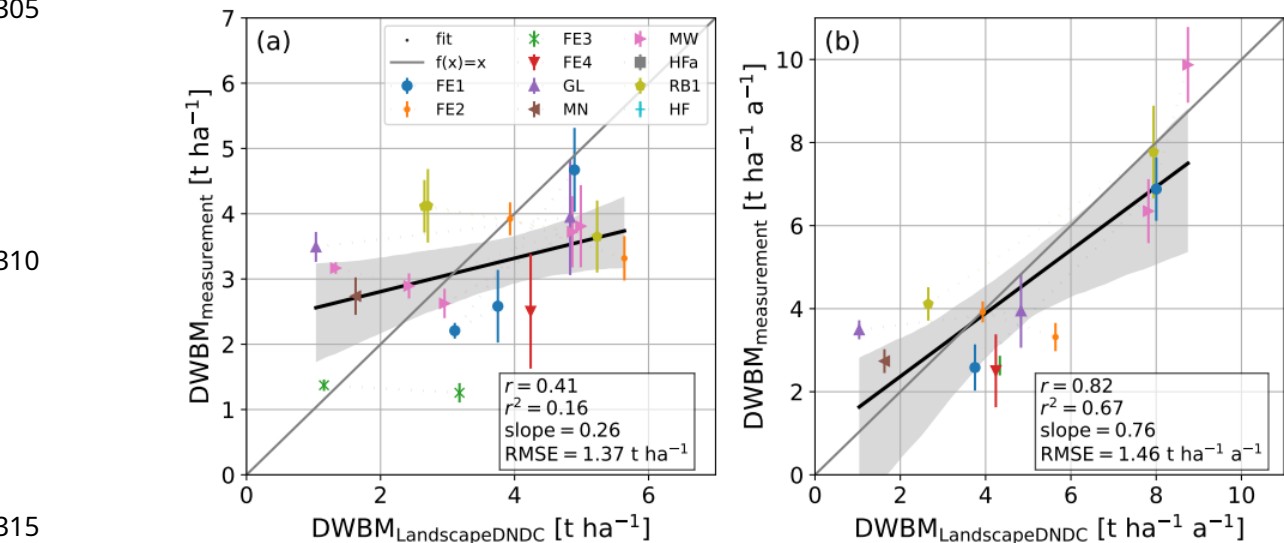

**Figure 3: Standing DWBM (mean values ±SDs) from one week before a cut measured in the RS campaign 2019 and 2020 against according values from LandscapeDNDC. (a) All data separately. (b) Data aggregated into yearly sums per plot.**

### 3.3 Climatic conditions

The MATs and MAPs weighted by field area and aggregated into hexagons for 2020 are shown in Fig. 4a and b, respectively, representing a climatically relatively normal year which is used as reference. The differences to 2020 are illustrated in Fig. 4c and d for 2018 and in Fig. 4e and f for 2019. The according mean values (±SD) of MAT and MAP over all studied fields are 9.4±0.4, 8.9±0.4, and 9.0±0.3 °C and 1002±110, 1175±196, and 1186±150 mm for 2018, 2019, and 2020, respectively. Hence, 2018 was about half a degree warmer and had about 180mm less precipitation than the other two years. The long-term mean values (2005–2020) of MAT and MAP are 8.4±0.3 °C and 1192±133 mm. As shown in Fig. 4, the deviations of MAT and MAP between the years did not occur spatially homogeneous, e.g. with temperature and rainfall anomalies in 2018 most strongly observed for the center and west of the study region, respectively.




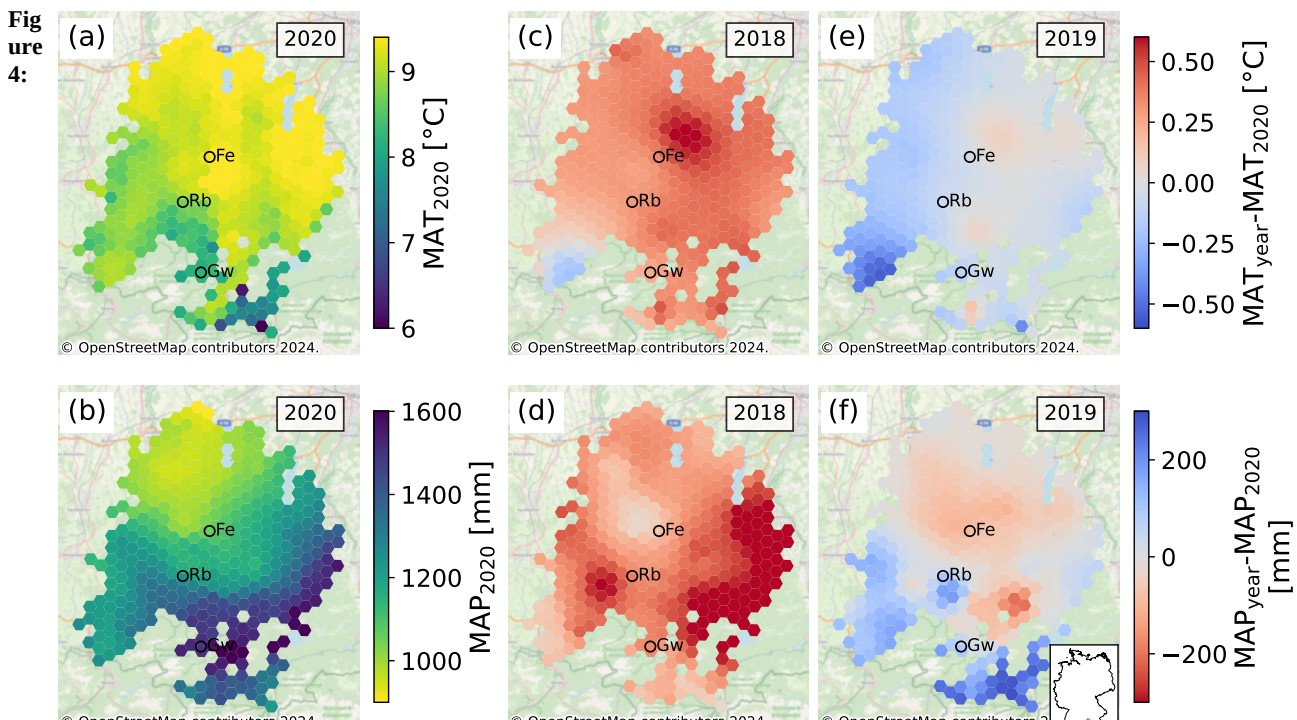

**330** **Average MAT (a) and MAP (b) for 2020 weighted by field area of all studied fields in the Ammer catchment. In c-f, the differences to the MAT (c, e) and MAP (d, f) in 2020 are shown for 2018 (c, d) and 2019 (e, f). The circles represent the location of validation sites Fe (Fendt), Rb (Rottenbuch), and Gw (Graswang). The location of the study region is illustrated by the inset in (f). Full copyright statement of background maps: © OpenStreetMap contributors 2024. Distributed under the Open Data Commons Open Database License (ODbL) v1.0.**

**335**

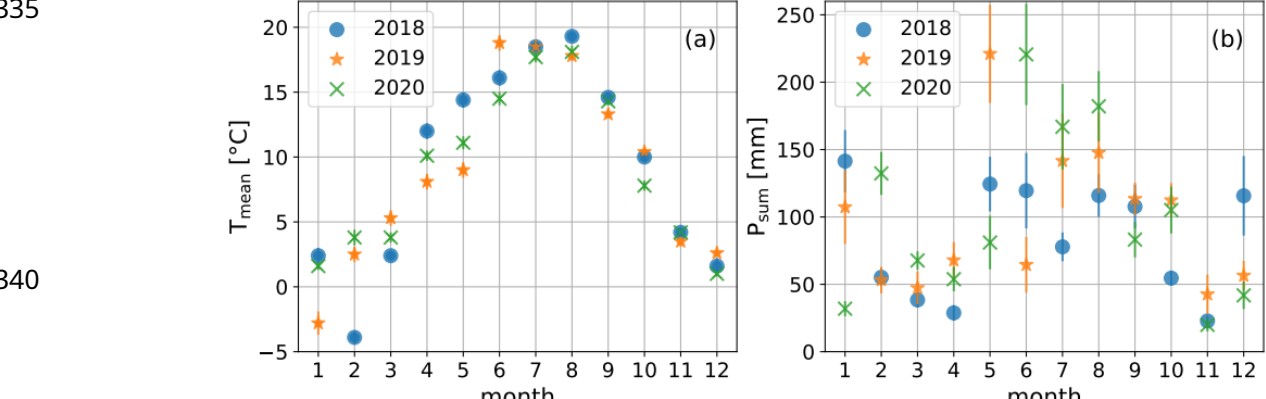

**Figure 5: Monthly mean (±SD) air temperature (a) and summed precipitation (b) weighted by field area of all studied fields in the Ammer catchment for the years 2018, 2019, and 2020.**

**345** For plant growth, in addition to annual weather conditions also the seasonal patterns of air temperature and precipitation matter. The monthly mean temperatures (a) and sums of precipitation (b) averaged over all fields are shown in Fig. 5. In 2018, the early spring precipitation (March and April) was exceptional low, with only 67 mm, compared to 115 mm and 121





mm in 2019 and 2020, respectively. Similarly, precipitation amounts in July and August were lowest in 2018. Mean monthly air temperatures in 2018 were below average in March, whereas they were far above average in April, May, July, August,

and September with up to 2.9 ° higher mean temperature in May. So overall, the growing season of 2018 can be characterized as drier and warmer than in 2019 and 2020.

The year 2019 had colder temperatures in April and May than the other years and was comparable warm like 2018 in June and July, however receiving about average (2020) amounts of precipitation. The year 2020 was characterized by average amounts of precipitation and mean air temperatures in spring, and by a wet summer (June, July, August) with below average

temperatures. Generally, across all years the SDs for monthly sums of precipitations are much higher than for monthly mean air temperatures, which means that the spatial variability of precipitation is stronger than for air temperature. As follows in Fig. 4, the air temperature range is relatively small for the vast majority of hexagons with a few colder areas in the south at higher elevations.

## 3.4 Simulated yields for 2018–2020

The simulated total regional yields as DWBM in 2018, 2019, and 2020 are 430 kt, 430 kt, and 450 kt, respectively. This corresponds to mean values of $7.51\pm1.53$ t ha$^{-1}$, $7.52\pm1.48$ t ha$^{-1}$, and $7.86\pm1.62$ t ha$^{-1}$. Besides the change in climatic conditions between the years, also the management differs for most fields (compare Fig. 2). In the drought year 2018, a regional mean yield decrease by 4.4 % in comparison to 2020 occurred, while for 2019, the mean yield reduction was 4.5 %.

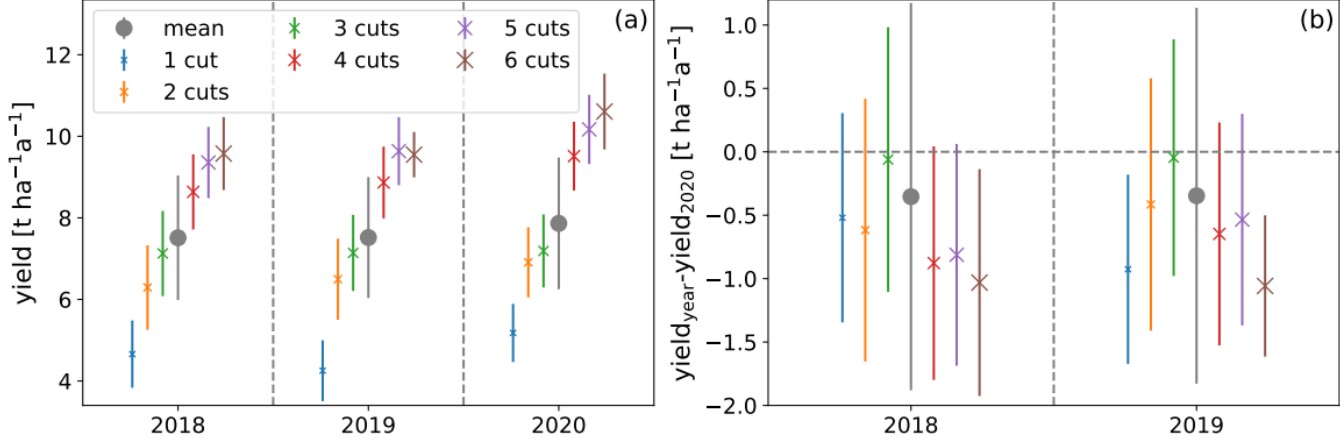

**Figure 6: Area means (±SDs) of harvested DWBM in different years for different numbers of cuts as well as averaged over all**
**fields (a) and difference to values in 2020 (b).**

The harvested biomass per area for different cutting regimes is illustrated in Fig. 6a. The relation between the number of cutting and manuring events to yield is strongly pronounced. Mean yearly harvest increases with management intensity with the exception of 6-cut fields in 2019. The SDs for more extensively managed fields are larger than for intensively managed fields. The minimum yearly harvest per area on a single field in the three studied years is 3.0 t ha$^{-1}$ a$^{-1}$ for 3-cut fields and 5.7

t ha$^{-1}$ a$^{-1}$ for 5-cut fields. They both occur in 2018. The according maximum values are 11.5 t ha$^{-1}$ a$^{-1}$ for 3-cut and 12.7 t ha$^{-1}$ a$^{-}$





[1] for 5-cut fields in the years 2018 and 2020, respectively. If one considers the differences to the mean harvest per area in 2020, shown in Fig. 6b, the negative trends are present for all cutting regimes. The decrease is statistically significant (p<0.01) for all managements. In 2018, the largest absolute mean decrease in yield is found for 6-cut fields, followed by 4-, 5-, 2-, 1-, and 3-cut fields, hence the yield decline tends to be higher for more intensively managed grasslands. Further, the

mean yield decrease in 2018 and 2019 for 3-cut fields is smallest.

Maps of the simulated yields in 2020 for 3- and 5-cut fields aggregated into hexagons are shown in Fig. 7. For 3-cut fields, the yields at higher elevations, in the southwest of the study region, are below average, whereas yields in the center of the study region are above average. This central region is also very productive under intensive (5 cuts) management. We note that mostly the low-yielding areas under 3 cuts, e.g. in the far South are not used intensively at all.


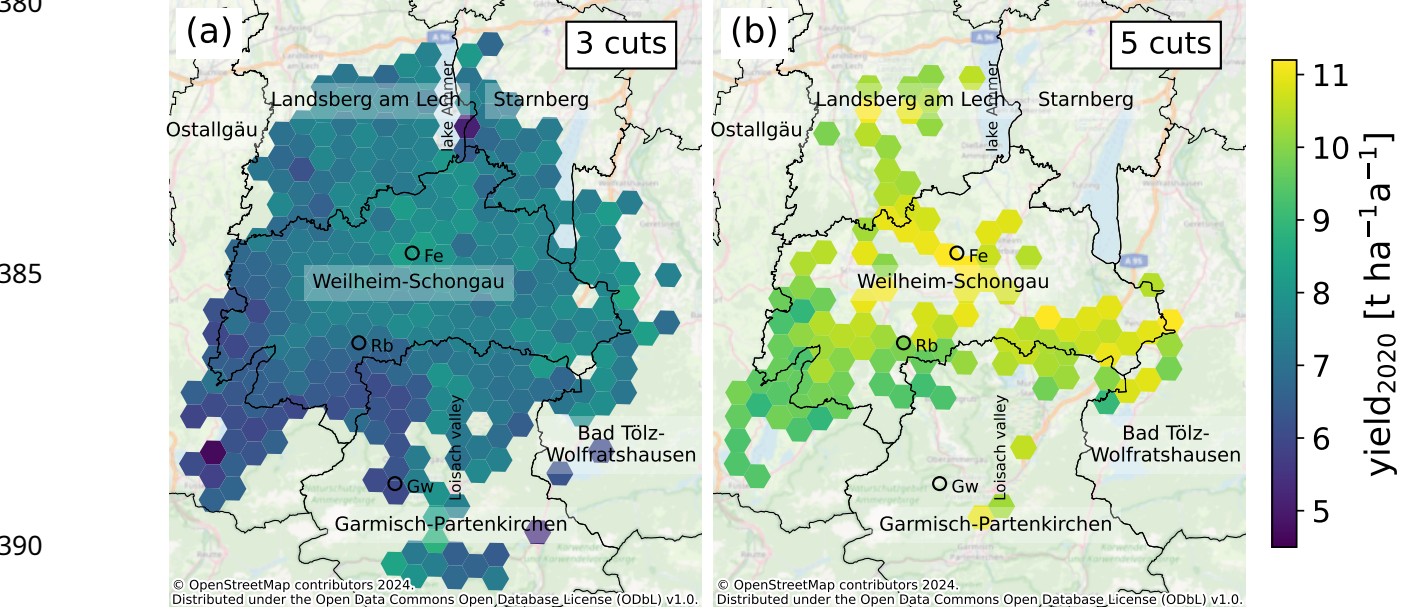

**Figure 7: Maps of the harvested DWBM of 3- (a) and 5-cut (b) fields in 2020 aggregated into hexagons.**

The differences between the yearly harvest in a hexagon for 3 (a, b) or 5 cuts (c, d) in 2018 and 2019 and the according value in 2020 are shown spatially resolved in Fig. 8. For 3 cuts, in 2018, 56 % of hexagons (161) have harvests smaller than in

2020. The hexagons with higher harvests than in 2020 are mostly located in the southwestern part of the study region close to the Alps (southern part of Ostallgäu) and in the Loisach valley. In 2019, a few more hexagons show yields below the ones from 2020 (173, 60 %). The regional mean reduction in 2019 is slightly higher than in 2018 (-0.09 t ha$^{-1}$ a$^{-1}$ vs. -0.05 t ha$^{-1}$ a$^{-1}$, Fig. 6). As areas with particularly small yields in 2019, we identify the Loisach valley and fields located in the district Bad Tölz-Wolfratshausen.

As in the study region less fields are cut 5 times than 3 times (Fig. 1), also fewer hexagons are plotted in Fig. 7c and d. There are also less fields per hexagon for 5 cuts (compare Sect. 2.4). For 5 cuts in 2018, the yields were lower than in 2020 in



almost all hexagons (82, 99 %). The mean reduction is 1.05 t ha$^{-1}$ a$^{-1}$. For 2019, the results are very similar with reduced yields in 96 % of hexagons and a mean reduction of -0.96 t ha$^{-1}$ a$^{-1}$. The main difference compared to 2018 is that in 2019 overall less fields were cut 5 times.





**Figure 8: Maps for 3-cut (a, b) and 5-cut (c, d) fields, where the color illustrates the difference between harvested DWBM in the**
**years 2018 (a, c) and 2019 (b, d) and the according values in 2020 both aggregated into hexagons.**

### 3.5 Correlation and regression analysis

At first, we use all 3 years of data on a field basis and include top soil properties, i.e., BD, SOC content, pH-value, and plant available water, elevation, MAT and MAP averaged over 2018–2020, as well as the number of cuts as potential predictors for annual yields. Even though the soil properties and the elevation remain the same for all years, the management on most

fields varies between the years (number of cuts and day of year with cut), thus the inclusion of all three years increases the number of soil and climate pairs per type of management.



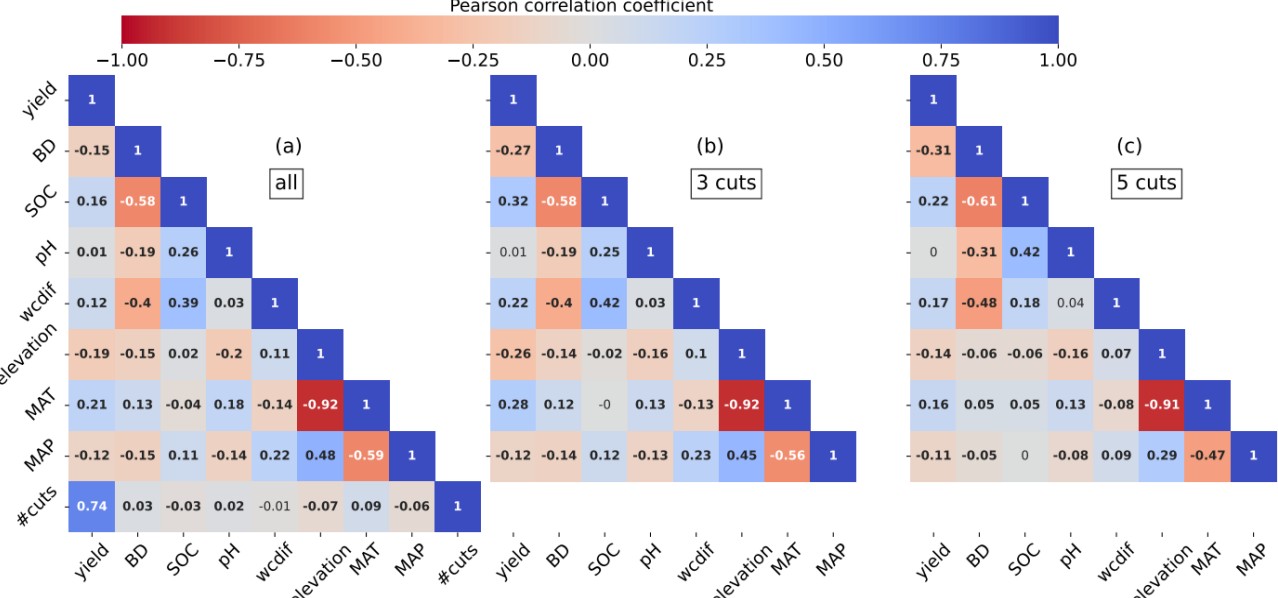

**Figure 9: Correlation matrices for field data linking soil properties (BD, SOC, pH, plant available water [wcdif]), elevation, climatic parameters (MAT, MAP) averaged over 2018–2020, as well as the number of cuts in the year (#cuts) with yearly harvested biomass (yields). Coefficients written (not) in bold indicate a p-value smaller (equal or larger) 0.01. (a) All fields, (b) 3-cut fields, and (c) 5-cut fields are considered.**

The correlation matrices for all fields, fields with 3 cuts, and fields with 5 cuts are shown in Fig. 9. If all fields are considered, the main correlation for the annual yield is given by the number of cuts with a Pearson correlation coefficient of 0.74 (CI=[0.735,0.741]). The regression analysis gives a slope of 1.2 t ha$^{-1}$ a$^{-1}$ per additional cut in a year. The coefficient of determination is 0.55, hence 55 % of the variation in yearly harvest can be explained by the number of cuts and the related number of manuring events.

If only fields with 3 cuts are analyzed, the next most important (anti-)correlations are given by SOC with a coefficient of 0.32 (CI=[0.311,0.331], slope of 0.17 t ha$^{-1}$ a$^{-1}$ per % increase in SOC) and the MAT with 0.28 (CI=[0.267,0.287], slope of 0.668 t ha$^{-1}$ a$^{-1}$ per °C), which is linked to the elevation (CI=[-0.27,-0.25], slope of -0.29 t ha$^{-1}$ a$^{-1}$ per 100 m). The SOC is anti-correlated with the BD and correlated with the nitrogen content of the soil (correlation coefficient 0.97 for all fields, not shown), as well as with the plant available water (correlation coefficient 0.39 for all fields). The elevation is strongly related to temperature with a correlation coefficient of -0.92. For fields with 5 cuts, the correlating parameters are the same but with lower values of r. As for 3 cuts, SOC is correlated the strongest with the yearly harvest with a coefficient of 0.22 (CI=[0.195,0.254], slope of 0.13 t ha$^{-1}$ a$^{-1}$ per % increase in SOC). SOC is anti-correlated with the BD. The (anti-)correlations with the plant available water, elevation, mean temperature, and precipitation are smaller with coefficients of 0.17 (CI=[0.137,0.197]), -0.14 (CI=[-0.171,-0.111]), 0.16 (CI=[0.128,0.189]), and -0.11 (CI=[-0.137,-0.076]), respectively.





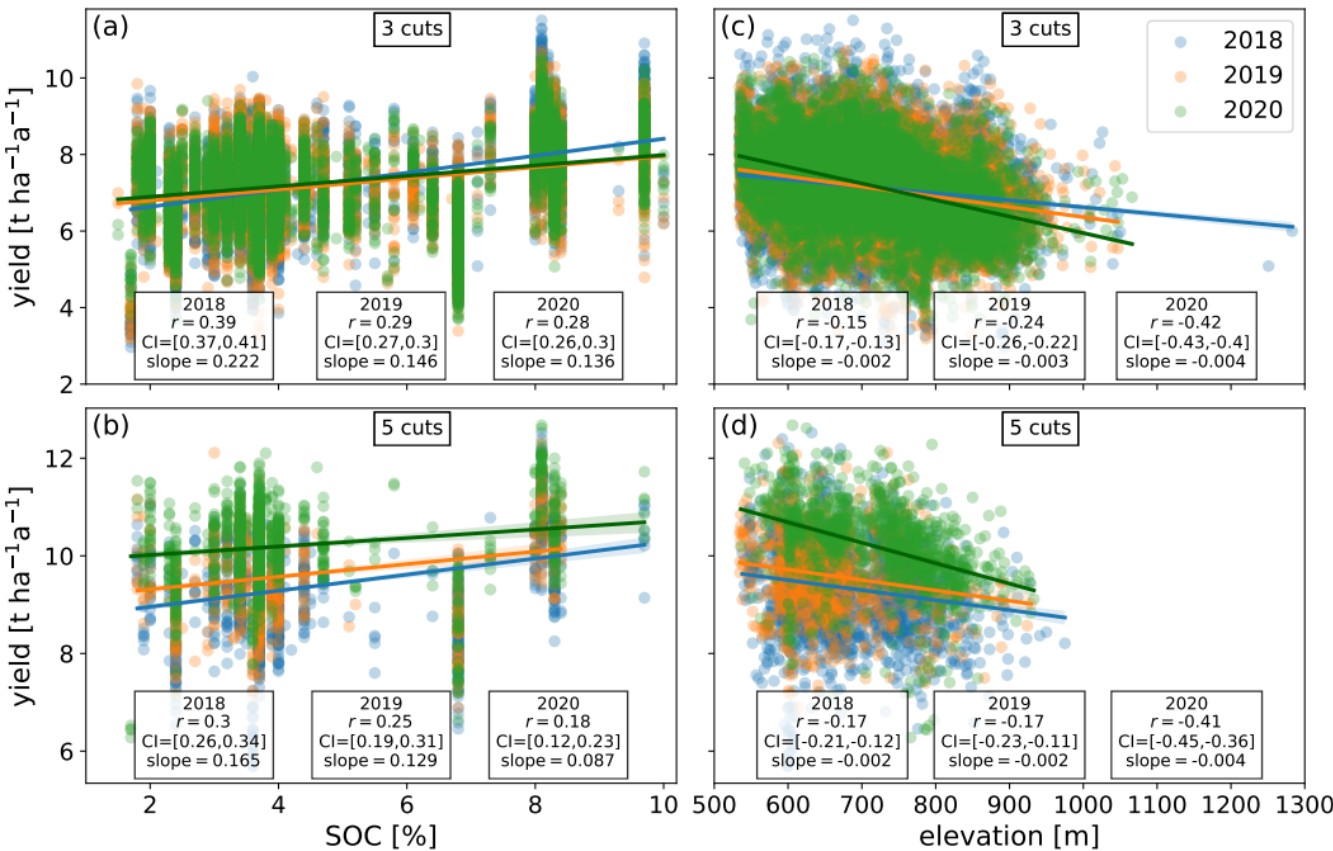

**Figure 10: Yields as DWBM in individual years split into management with 3 (a, c) and 5 cuts (b, d) plotted against SOC (a, b) and elevation (c, d). Every point represents a single field-year pair. Fitted regression and associated correlation coefficients and slopes are given in the boxes for each year. All regressions are significant (p<0.0005).**

Secondly, we evaluate the correlations within individual years, in particular, to contrast the drought year 2018 and the rather normal climatic year 2020. If all fields in a year are considered, again the correlation between the number of cuts and yearly harvest is dominant (not shown), as expected. The correlation coefficients range from 0.73 (2018) to 0.78 (2020). The correlation between SOC and yield is the strongest in 2018 with a coefficient of 0.21 (CI= [0.20,0.22], slope of 0.17 t ha$^{-1}$ a$^{-1}$ per % increase in SOC) compared to the other years with r=0.16 (CI=[0.14,0.17], 2019) and r=0.11 (CI=[0.10,0.12], 2020), which also show smaller slopes in the regression. If only fields with a certain management are included in the analysis, the remaining correlations generally become stronger. The regressions for all fields with 3 or 5 cuts against SOC and elevation for individual years are shown in Fig. 10 including correlation coefficients and associated CIs. The correlation between SOC and yearly harvest as well as the slope in regression are stronger in 2018 (r=0.39, slope of 0.22 t ha$^{-1}$ a$^{-1}$ per % increase in SOC) than in 2019 and 2020 for 3-cut fields. For 5-cut fields, the correlation is also the strongest in 2018 (r=0.30), decreases for 2019 (r=0.25), and drops further in 2020 (r=0.18). Note, that some fields are linked to the same soil profile (compare Sect. 2.3.1), which is why the yields in Fig. 10 on the left partly lie on vertical lines.



Considering the correlation between elevation and yearly harvest, all correlation coefficients are negative. For 3-cut fields, the correlation coefficients are -0.15, -0.24, and -0.42 for 2018, 2019, and 2020, respectively. For 5-cut fields the coefficients are -0.17 in 2018 and 2019, and -0.41 in 2020. The according slopes behave similar. The slopes suggest a decrease between 0.2 t ha$^{-1}$ a$^{-1}$ per 100 m elevation increase in 2018 and 0.4 t ha$^{-1}$ a$^{-1}$ per 100 m elevation increase in 2020. Further, the elevation of a field can explain more of the predicted yield in a normal year than in a dry year.

## 4 Discussion

### 4.1 Ability to reproduce grassland yields in the study area

The model performance measures (r² and RMSE) for grassland yield simulations derived from single cutting event comparisons are among the best values reported in other studies (Lazzarotto et al., 2010; Liebermann et al., 2020; Petersen et al., 2021; Sándor et al., 2017). In addition to previous work (Petersen et al., 2021), in this study LandscapeDNDC was further validated for another intensive site (Rottenbuch), and for the first time also for yields of extensive grassland management. In LandscapeDNDC, both management intensities are currently represented within a global species parametrization. We believe that a distinction between plant functional types, which could highly differ between different management intensities, has the potential to further improve the model performance at field scale. However, on the regional scale, spatial explicit information on plant community composition is lacking.

When considering individual cutting events, the performance measures obtained in the uncertainty assessment with regional data are worse than those obtained in the validation. Firstly, this is related to the measurements. While for the yield validation data 3 m² of biomass were cut, only 1 m² was cut for the regional validation data which were located at different positions in the field. There can be considerable variability in soil properties at the field scale, which cannot be captured by a single soil parametrization as used for model input. The evaluation measurements are therefore more prone to uncertainties, which can also be seen by comparing the SDs in Fig. 2 and Fig. 3. Secondly, this can be related to the uncertainty of the model input data, which are derived from on-site measurements for the validation, whereas they are derived from regional soil databases for the regional yield uncertainty assessment. Some data, such as the rates and timing of manure application, are only estimated for the regional evaluation, while they are known exactly for the simulation of the validation data. We expect the fit for individual cuts to improve in the regional evaluation, if measured on-site information were available. However, the uncertainties in inputs balance out when considering the aggregated output, as long as the inputs are not systematically biased. For instance, some fields may receive less fertilizer than in reality, whereas others receive more. On average, the harvested biomass stabilizes around the true value (if deviations from real values remain within a small range in which linear approximations for relevant processes perform) as long as there are no biases in the input data, such as systematically underestimated amounts of fertilizer. Similarly, the aggregation of single cutting events to yearly total yields may compensate for the underestimation of small DWBM and the overestimation of large DWBM by LandscapeDNDC (Fig. 3), as individual yields typically decrease with each cutting event over the course of a year (Fig. 2). For these reasons,





the performance measures of the aggregated data are much better than the individual measurements, with an $r^2$ of 0.67 and a slope of 0.76 (Fig. 3). The latter means, that for the yearly harvests from individual fields, large yields tend to be overestimated and small yields tend to be rather underestimated. The deviation between simulated and measured DWBM before actual cutting events is 6.9 % in the regional uncertainty evaluation and we therefore expect LandscapeDNDC to slightly underestimate total regional yields.

At regional level, the same uncertainties apply and the performance can be expected to be similar to the regional evaluation. The highest accuracy is achieved for most aggregated (in space and time) results. Nevertheless, there are some biases in the input that are worth mentioning. Firstly, the number of cuts is slightly underestimated, since RS may miss some cutting events due to cloud cover (Reinermann et al., 2022). Secondly, the total regionally applied amount of slurry depends on the number of cuts, which was higher in 2018 than in 2020 (mean of 3.21 vs. 3.07), and resulted in an average of 6 % higher N-input in 2018 compared to 2020 in the simulations. In reality, it is more likely though that the amount of slurry available in each year is rather constant given more or less static cattle numbers. Therefore, the manure input as chosen in the present simulations likely leads to an overestimation of the yields in 2018 as compared to 2020. Thirdly, fields on SOC-rich soils (SOC > 10 % in top soil), and therefore likely very productive fields with large nitrogen contents, are excluded.

Overall, the results of the validation and regional uncertainty evaluation are very convincing, with deviations of 3 to 10 % between modeled and measured yields aggregated over many years. The uncertainty of regional yields should be similarly small. The uncertainties of yields aggregated into hexagons (Fig. 7 and Fig. 8), may be higher if only a small number of fields are considered for aggregation.

## 4.2 Impact of management intensity on yields

Previous studies have revealed that grassland yields are mainly controlled by management intensity (Dellar et al., 2019; Grigulis and Lavorel, 2020; Schlingmann et al., 2020; Wang et al., 2021). However, in our study region the number of cuts at a given single field was observed to vary between years (Fig. 1). Therefore, it is not trivial to associate a uniform management intensity to a field over many years. To increase the sample size and cover the whole study region, we therefore considered field-year pairs as independent data points, and analyzed spatially aggregated (i.e. hexagon) data rather than single fields. To the best of our knowledge, the present study is the first to employ such high-resolution cutting input data from RS in regional grassland simulations.

From all considered influential factors on yields, the number of cuts and associated manure applications, and therefore the management intensity, was found to exhibit the largest correlation, while yield was not strongly correlated to any soil and climate properties (Fig. 9). Strong relation between yields and management has also been found in previous experimental (Grigulis and Lavorel, 2020; Schlingmann et al., 2020; Wang et al., 2021) as well as modeling studies (Dellar et al., 2019). With our approach, 55 % of variation in yields can be explained by the number of cutting events and associated manure application with an average yield increase of 1.2 t ha$^{-1}$ a$^{-1}$ per additional cut. The importance of the management is also reflected by empirical yield models using management intensity as a parameter (Huguenin-Elie et al., 2017). A regression





model based on multiple experiments throughout Europe and including weather, altitude, and management as parameters found a slope of 1.3 t ha$^{-1}$ a$^{-1}$ per additional cut (Dellar et al., 2019). The Bavarian state agency for agriculture (Landesamt für Landwirtschaft [LfL]), reports for meadows, mean yield estimates of 3.4 t ha$^{-1}$ a$^{-1}$, 4.7 t ha$^{-1}$ a$^{-1}$, 6.8 t ha$^{-1}$ a$^{-1}$, 7.7 t ha$^{-1}$ a$^{-1}$, 9.4 t ha$^{-1}$ a$^{-1}$, and 10.2 t ha$^{-1}$ a$^{-1}$ for 1-6 cuts, respectively, and for mowing-pastures a range of 5.7 to 9.4 t ha$^{-1}$ a$^{-1}$ (Bayerisches

Landesamt für Landwirtschaft, 2018). Our simulations resulted in 5.2 t ha$^{-1}$ a$^{-1}$, 7.0 t ha$^{-1}$ a$^{-1}$, 7.4 t ha$^{-1}$ a$^{-1}$, 9.6 t ha$^{-1}$ a$^{-1}$, 10.3 t ha$^{-1}$ a$^{-1}$, and 10.7 t ha$^{-1}$ a$^{-1}$ for meadows and a range of 5.1 to 10.0 t ha$^{-1}$ a$^{-1}$ for mowing pastures in the wider Ammer catchment for 2020, a year with average weather conditions. These values compare well with the regional yield estimates of LfL and show the same trend, but with slightly higher values for simulated low-intensity meadows.

We note that at regional scale extensive management is associated with greater absolute variation in yields than intensive

management, as can be seen from the SDs in Fig. 6. This is due to the fact that exceptionally small or large yields from an individual cut have a bigger weight in the yearly cumulated harvest on a field with fewer cuts than on a field with many cuts.

## 4.3 Impact of climate conditions on regional grassland yields

For meadows and mowing-pastures in the larger Ammer catchment (583 km²), we found total yields of 430 kt, 430 kt, and 450 kt, for the years 2018, 2019, and 2020, respectively. Per area, mean yields were 7.51 t ha$^{-1}$, 7.52 t ha$^{-1}$, and 7.86 t ha$^{-1}$.

This relates to a yield decrease of 4 % in 2018 and 2019 compared to 2020 (MAT 9.0 °C, MAP 1186 mm), which can be explained by the drought conditions in 2018 (9.4 °C, 1002 mm) and associated effects also in 2019 (8.9 °C, 1175 mm), compare Sect. 3.3. Generally, spatial and temporal development of grassland yields are not well tracked, thus there are only few coarse references for comparison available. From the Bavarian state agency for statistics, annual yield statistics based on expert estimates per district and year for meadows are provided (Bayerisches Landesamt für Statistik, 2024). In the district

Weilheim-Schongau, in the center of the study region, mean yields of 7.07 t ha$^{-1}$, 7.93 t ha$^{-1}$, and 8.25 t ha$^{-1}$ for 2018, 2019, and 2020 are given, respectively. The according simulation results (compare Fig. 7 and Fig. 8) are 7.99 t ha$^{-1}$, 8.02 t ha$^{-1}$, and 8.26 t ha$^{-1}$. The simulated yields show the same trend between 2018 and 2020, but are slightly higher than the values reported from the authority. The largest deviation is found for 2018 with nearly 1 t ha$^{-1}$ a$^{-1}$ higher yields from the simulations. The differences between the years in the data from the state agency are larger than for the simulation results. For meadows in

Weilheim-Schongau, a reduction of 14 % in 2018 compared to 2020 is given, whereas the simulations show a reduction of only 3 %. The district of Garmisch-Partenkirchen is mostly included in the study region. However, only data for 2018, 6.25 t ha$^{-1}$, is available from statistics. The LandscapeDNDC simulations again resulted in larger yields of 7.42 t ha$^{-1}$. All other districts are only partly overlapping with the study region, thus the comparison is less adequate. For Ostallgäu (578-2082 m a.s.l., mean: 832 m a.s.l.) and Landsberg am Lech (521-853 m a.sl., mean: 625 m a.s.l.), the official statistics give the

maximum yield of the three years for 2018, whereas we find maximum yields in 2020, compare Fig. 8.

In the simulations, annual yields differ spatially with some positive responses to climatic conditions observed in 2018 for 3-cut fields, and mostly negative ones, e.g. in the north-east of Weilheim-Schongau (lower warmer climate) and negative ones for almost all 5-cut fields. This variation is in line with experimental findings from Switzerland, where three grassland sites





were equipped with rain-out shelters (MAP reduction of 30 %) (Finger et al., 2013), which lead to yield decreases of roughly

25 % for an intensively managed wet pre-alpine (393 m) and an extensive alpine site (1978 m), but to about 10 % yield

increases for a wet pre-alpine site (982 m) managed at low intensity (Finger et al., 2013). Similarly, a RS study analyzed the

leaf area index of grasslands of several mountain farms in north-eastern Italy (Castelli et al., 2023) and found positive as well

as strongly negative effects on different parcels for the drought summer 2022. Regarding, the decrease in yields in 2018, an

evaluation of the enhanced vegetation index [EVI] – an indicator for vegetation health and productivity - found above

average values (reference period 2000-2018) in spring 2018 within all districts overlapping the study region, whereas in

summer it was below average, where data was available (Reinermann et al., 2019). This suggests at least normal yield

amounts from the first and in terms of the contribution to annual yields, the most important cut. Experimentally, yields from

the first cut (yearly yields) from the lysimeters in Fendt, Rottenbuch, and Graswang were decreased in 2018 compared to

2012-2018 by 17 % (18 %), 28 % (12 %), and 10 % (3 %), respectively (Petersen et al., 2021). The years 2018 to 2020 were

also evaluated in regard of drought periods and impacts on yields with lysimeters from an Austrian permanent grassland at

conditions similar to our study (Forstner et al., 2023). Drought spells were found in spring and late summer of 2018 causing

diverse effects on yields, whereas the more extreme summer was found to be in 2019 leading to severe yield losses. In the

present study, the yields in 2019 were found to be slightly smaller than in 2018. This may be linked not only to climatic

conditions but also to a reduced amount of plant storage built up at the end of 2018 (Addy et al., 2022; Chapin III et al.,

1990), impacting the most important first cut in the year (Fig. 2), which was still average in 2018. Another reasons for rather

low yields in 2019 is the spatially uneven distribution of precipitation (compare Fig. 4).

Further, we note increased nitrous oxide emissions in drought impacted years with reduced yields and thus lower plant

nitrogen uptake, while nitrate leaching rates remain relatively unchanged (Fig. A4 and A5).

From simulations of climate change scenarios including raised $CO_2$ for intensive management within the Ammer catchment

on a site-scale, yield reductions of 15 % were found for drought years with less than 550mm between March and October

(Petersen et al., 2021), which is an even bigger drop in precipitation than in 2018 (Fig. 5). The direct comparison of all

individual fields with the same number of cuts in 2018 and 2020 (n=9777) from our regional simulations leads to a mean

yield reduction of -6±12 % (-11±7 % if only fields with yield reduction are considered), which is in the range of previous

site-scale findings.

The field-based simulations performed in this study explicitly included many pedo-climatic conditions and different

grassland management regimes, which resulted in the large spatial variation of yields in the study area (Fig. 7 and Fig. 10).

This allowed for the analysis of versatile responses in different climatic conditions (MAP and MAT as function of elevation)

in the study region. The correlation analysis revealed decreasing yields with increasing elevation (-0.4 t ha$^{-1}$ a$^{-1}$ per 100 m),

and that this effect is weaker in the drought year 2018 (-0.2 t ha$^{-1}$ a$^{-1}$ per 100 m, Fig. 10). This is plausible, since plant growth

in pre-Alpine areas is mainly energy limited (Forstner et al., 2023) and under normal conditions productivity decreases with

elevation, as is also employed in empirical yield estimates (Dellar et al., 2019; Huguenin-Elie et al., 2017). Our results

suggest that the energy limitation at higher altitudes weakens in an above average warm year like 2018, whereas in a colder



year like 2020 (below average temperature in May-August, Fig. 5) harvests of fields at higher altitudes cannot reach the amounts of those at lower altitudes. And, the other way around, that dry conditions at lower altitudes become limiting under

climatic conditions as present in 2018 as also presented by Forstner et al. (2023).

Climatic conditions also effect soil processes like mineralization, which is particularly important for plant growth in grassland in terms of nitrogen supply (Schlingmann et al., 2020). On the one hand, warming has been found to increase mineralization in a translocation experiment along an elevational gradient (Wang et al., 2016). On the other hand, it is known, that reduced soil water contents lead to decreased mineralization rates (Emmett et al., 2004; Larsen et al., 2011) and

therefore less plant available nitrogen, which is particularly important in grassland systems as Schlingmann et al. (2022) found higher [15]N recovery rates of labeled slurry in the soil N-pool than in plants. These mineralization effects on yields are intrinsically included in our process-based simulations and can be linked to the regression analysis between yields and SOC (Fig. 9), where we found a significant positive correlation. It is plausible, that yields are generally raised on fields with higher SOC indicating also higher nitrogen availability, plant available water, and therefore larger mineralization rates, in

particular under dry conditions. The enhanced growth on SOC-rich soils and the buffering of weather persistence, was also discovered in an experimental study (Reynaert et al., 2024), where two contrasting soils were used. Our regression analysis of individual years revealed yields to increase with a slope of 0.09 t ha$^{-1}$ a$^{-1}$ per % increase in SOC in the rather normal year 2020, and with a slope of 0.17 t ha$^{-1}$ a$^{-1}$ per % increase in SOC in the drought year 2018.

Note, that correlating the yield differences between years per field is only possible for a reduced number of fields because of

the management variations across single fields and years (Fig. 1). For 3-cut fields, the study region is well represented and statistically significant results occur. The same trends, i.e. smaller absolute yield losses at higher elevations and SOC are observed. Further, many of the processes relevant in grassland ecosystems are described by non-linear equations, hence linear regressions can only grasp major trends.

**5 Conclusions**

In this study, we showed that regional grassland yields drop in a drought year like 2018 even in a area with high MAP like the Ammer catchment exemplifying the northern pre-Alps. We applied a field-scale modeling approach for a larger grassland region, to the best of our knowledge, for the first-time using RS-derived cutting dates to determine grassland management inputs. The analyzed multitude of pedo-climatic conditions can neither be studied experimentally nor captured within empirical models and allowed the quantification of trends by regression analysis. Yields had the strongest correlation

with management intensity and linear regression revealed a yield increase of 1.2 t ha$^{-1}$ a$^{-1}$ per additional cut and associated fertilizer application. We further found yields to increase with SOC and decrease with elevation and, thus lower MAT. In a warm and dry year, the positive impact of SOC on yields is raised, whereas the temperature limitation at higher elevations is reduced.

In the face of climate change, we therefore expect management intensification at higher elevations and potentially on SOC-rich soils in particular in climatically extreme years to partly even out negative impacts from drought at lower elevations in this pre-Alpine region. The potential to reduce yield losses at higher SOC stocks under drought gives yet another reason (additional to $CO_2$ storage, water retention, etc.) for further investigation and application of SOC enhancing management practices.

### Data availability

The data are available from the corresponding author on reasonable request.

### Author contributions

CB and RK planned the work. CB produced regional model inputs, carried out LandscapeDNDC simulations and analysis of outputs. SR provided regional cutting dates. RL and RW provided climate data. AS provided regional biomass measurements for further model evaluation. DK provided technical and scientific model support and critically reviewed the validation and regional evaluation. RK conceived the original research idea and supervised the project. CB and RK led the writing of the manuscript, with contributions from all co-authors, who also critically reviewed the manuscript.

### Competing interests

The authors declare that they have no conflict of interest.

### Acknowledgments

Infrastructure for the research was provided by the TERENO Bavarian Alps/ Pre-Alps Observatory, funded by the Helmholtz Association and the Federal Ministry of Education and Research (BMBF), and funding was provided by the Federal Ministry of Education and Research (BMBF) via the BonaRes project SUSALPS (031B0027A). Furthermore, this work was also funded by the German Federal Ministry of Education and Research (BMBF) project "Integrated Greenhouse Gas Monitoring System for Germany – Sources & Sinks (ITMS Q&S)" under grant number 01 LK2105A. The authors would like to thank Josef Hammerl (LfU) for making available the Bavarian soil dataset and by supporting transformation into LandscapeDNDC model input.





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

**Appendices**

**A Measurement campaign used for uncertainty assessment**

Per site and measurement date, the vegetation in 4 plots (50x50 cm) was cut down to 7 cm height and collected. Afterwards, for one of the plots the vegetation was further cut down to 2 cm height and collected. The plot positions were chosen such, that no effects from previous measurements occurred. The vegetation was dried and the DWBM determined. For model validation, we took the mean values of the >7 cm DWBM and added the 2-7 cm DWBM scaled up linearly to 0-7 cm. A total number of 83 measured DWBM was reached. As part of the field campaign, on-site cameras were analyzed for cutting dates

and mowing periods.



## B Aggregated validation results

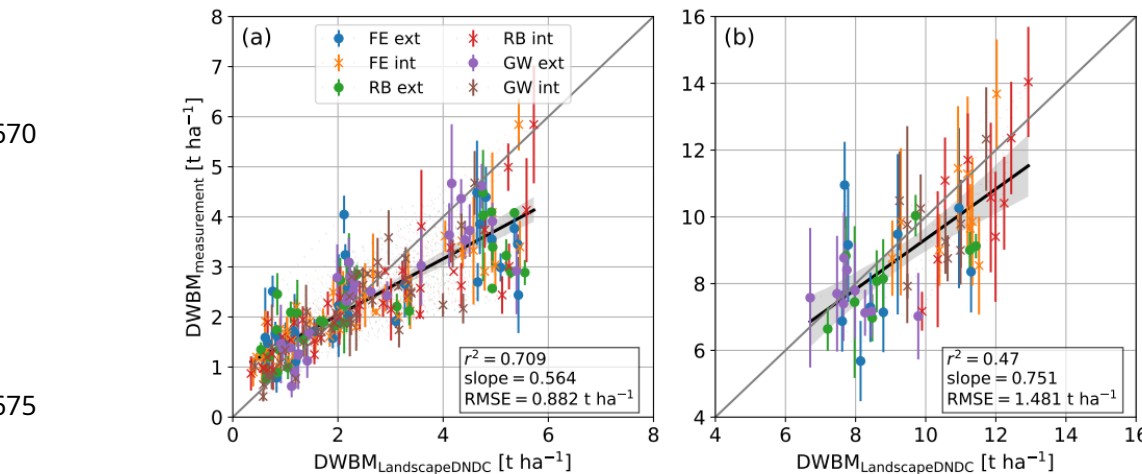

**Figure A1: Yields from the validation sites Fendt (FE), Rottenbuch (RB), and Graswang (GW) for extensive (ext) and intensive (int) management from LandscapeDNDC simulations against measured values (±SDs) till 2021, excluding 2013. (a) All cuts are shown individually. (b) Data is aggregated into yearly yields per site and management.**

## C Climatic conditions in spring

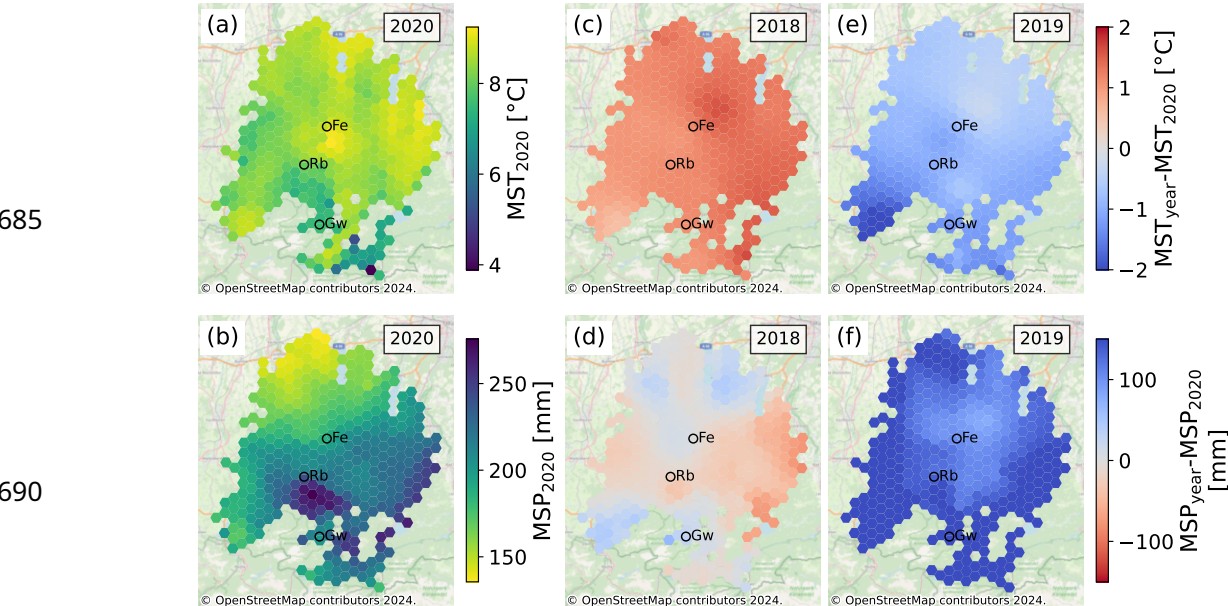

**Figure A2: Average mean spring (March, April, and May) temperature [MST] (a) and summed spring precipitation [MSP] (b) for 2020 weighted by field area of all studied fields in the Ammer catchment. The differences to 2020 are shown for 2018 (c, d) and 2019 (e, f). Full copyright statement of background maps: © OpenStreetMap contributors 2024. Distributed under the Open Data Commons Open Database License (ODbL) v1.0.**




## D Additional maps of simulated yields in 2018, 2019, and 2020

**Figure A3: Harvested DWBM illustrated by the color in maps for 3-cut (a, b, c) and 5-cut (d, e, f) fields in the years 2018 (a, d), 2019 (b, e), and 2020 (c, f) aggregated into hexagons.**




## E Regional nitrogen losses

In this study, we focused on yields, because of the availability of validation and evaluation data. Generally, LandscapeDNDC is well tested for simulating nitrogen fluxes. In this appendix, we therefore show the simulated $N_2O$-emissions and $NO_3$-leaching losses determined for grasslands in the larger Ammer catchment for 2018-2020.

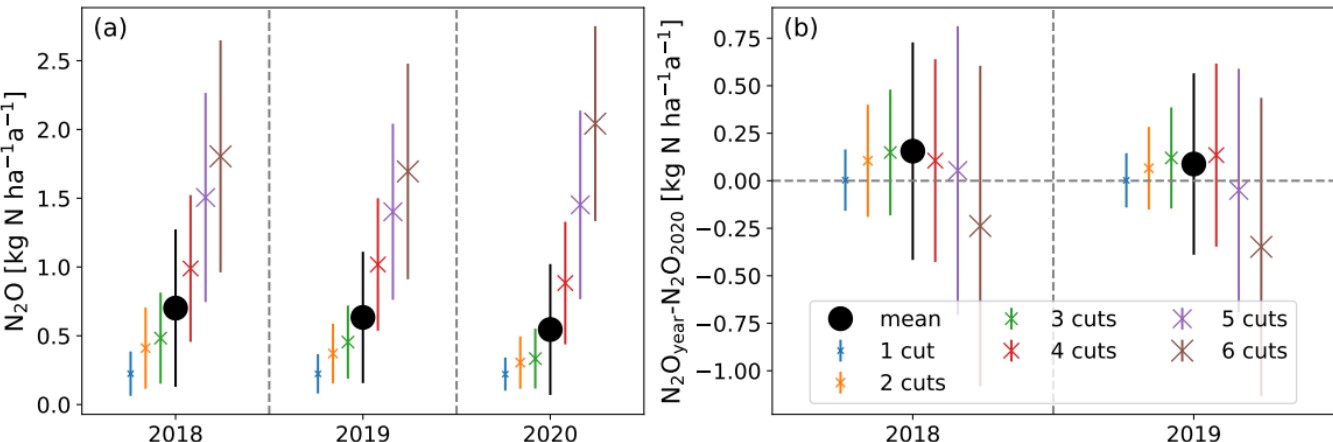

**Figure A4: Area means (±SDs) of nitrous oxide emissions in different years for different numbers of cuts as well as for all fields together (a) and difference to values in 2020 (b).**

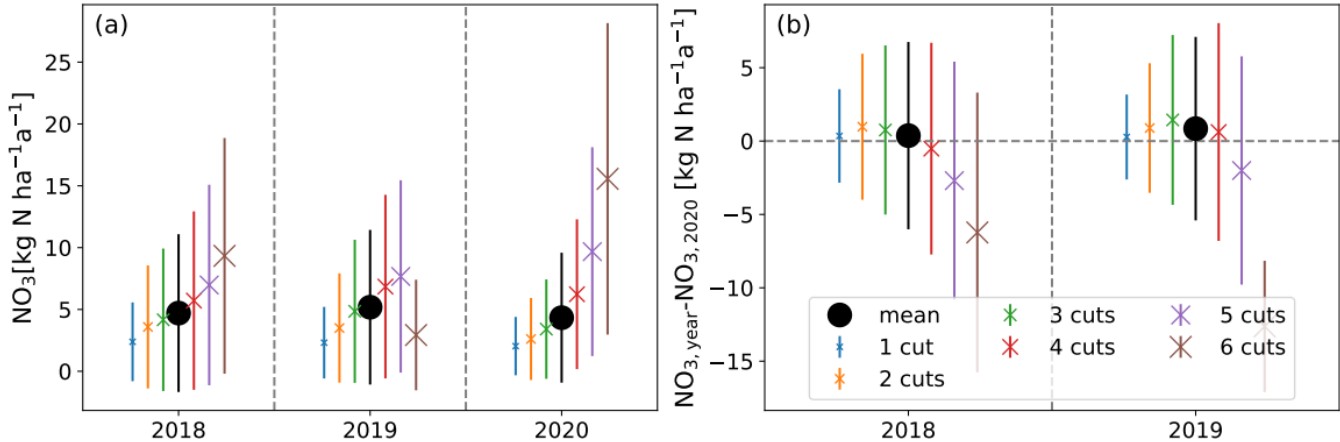

**Figure A5: Area means (±SDs) of nitrate leaching in different years for different numbers of cuts as well as for all fields together (a) and difference to values in 2020 (b).**