# Peer review of "Drought impact on productivity: Data informed process-based fieldscale modeling of a pre-Alpine grassland region"

_EGUsphere, 2024_

## Author Comment (AC1)

*The referee comments are upright, and our responses are in italics.*

**RC1:** The present study describes simulations of grassland yields in the pre-Alpine region of Southern Germany under different management conditions and one drought year. The study uses the established model LDNDC together with a wealth of local and regional data and concludes that management, in particular number of cuts impacts grass yield and that drought reduces biomass production.

I found the paper to be very narrow in focus and to rely heavily on knowledge of how LDNDC works and the particulars of grassland management in Bavaria. The title and aim sound very promising and the drought impact on grass yield is critical in multiple regions of the world. However, the paper goes on to discuss aspects like number of cuts in detail, with only a fraction of the analysis focusing on drought effects.

**Response:** *We recognize the comment that only a fraction of the analysis is focusing on drought effects and in a revised manuscript we will strengthen the focus on this point. We are planning to determine drought indices (SPI, SPIE, or similar) per field. These will be used to extract the periods of strongest drought, which can then be contrasted to the other periods. We will also include the drought indices in the correlation analysis and hope that this approach will pin-down more clearly, which effects stem from drought. The refined analysis will also feed into the abstract. We agree that the title does not fully match the study and suggest to change it to "The dual role of management and drought on productivity: Data informed process-based field-scale modeling of a pre-Alpine grassland region".*

*The narrower spatial focus of this paper was a deliberate choice to provide an in-depth analysis under actual, observed field management conditions. Such detailed data is usually not available at larger scales, which requires generalizations that can introduce considerable uncertainties. For the study region, we had access to uniquely detailed and reliable datasets. Firstly, the employed dates of cutting events stem from a remote sensing-based model, which was trained on data from the study region. This kind of data does not exist globally and to the best of our knowledge, this is the first study, where data derived from remote sensing was used to inform the grassland management of a biogeochemical model. Secondly, the soil data was derived together with the local authorities, which allowed us to simulate realistic soils with appropriate soil organic carbon stocks, whose spatial variation is not adequately adressed by national or EU/ global soil maps. We are not aware of a similar product on a larger scale. However, the quality of process-based simulations can only be high, if the input data is of high-quality, too. For instance, the effect of increasing yields with increasing soil organic carbon contents, and a strengthening of this effect under drought, could have not been extracted with a soil data base with a smaller range of soil organic carbon contents. Some effects, like the relation between yields and soil organic carbon could not be extracted with coarser soil input data. We want to emphasise that detailed inputs are required to empower process-based models.*

**RC1:** I would be interested to see a more in depth analysis of the variation in drought impact and whether management has any impact on how affected by drought yields are.

**Response:** *For the first point, see the answer above. For the second point, we also find this an important topic in particular regarding management adaption in a changing climate. This is partly discussed in Sec. 3.4. The model predicts the largest mean yield decrease for 6-cut fields, followed by 4-, and 5-cut fields, whereas the more extensively managed fields show smaller absolute yield*

*decreases (Fig. 6, and text below L371-375) which agrees well with experimental literature (e.g. Korell et al., 2024). We thank the referee for noting, that this point was not emphasised enough and in the revised manuscript, we are planning to deepen the analysis and discussion on this topic.*

**RC1:** It would also be useful to include a discussion of the generality and portability of this study, especially given that the study area must be one of the regions with the highest data availability in the world.

**Response:** *We fully understand the concern of the reviewer that the portability of the study is not discussed. We analysed an area of 4600 km² in the northern pre-Alps, which was chosen due to the exceptionally high accuracy of available input data. Even though the input data is not portable to other regions, the results are expected to hold for pre-Alpine areas in Switzerland, France, Italy, Austria, and Slovenia at comparable precipitation levels (800 - 1800mm), which are illustrated in Fig. 1, and potentially even other European mountain ranges. We expect that derived trends like increasing yields with increasing soil organic carbon content and buffering of negative impacts of drought on productivity are portable to these regions. In a revised version of the manuscript, we will include a discussion of the portability of the study and emphasise the unique quality of our input data, such that it becomes clear, that we balanced data requirements and possibilities that process-based models yield.*

[Figure]

*Figure 1: Mean annual precipitation sums in mm from 1971-2019 for the Alps and pre-Alps (Isotta et al., 2014). The findings from our study are expected to apply for Alpine regions with precipitation sums between 800 – 1800 mm per year.*

The underlying research questions around drought and the model simulations in themselves are a good fit for Biogeosciences but in the current form, the paper would find a better form in a more agricultural or even more regional journal.

**Response:** *Thanks for the comment. We believe that by revising the manuscript and a more in-depth analysis of drought influenced biogeochemical impacts on grassland yields and by strengthening impacts on nitrogen cycling, we hope that the paper will fit for Biogeosciences.*

**Specific comments**

**RC1:** Abstract the text seems to mainly discuss the effects of biomass harvest, with passing mention of drought effect, contrasting to the title.

**Response:** *We thank the referee for this comment. In a revised version of the manuscript (including drought indices), drought, and drought-biogeochemistry impacts will also be emphasised stronger in the abstract.*

**RC1:** Section 31, figure 2 It would be good to have an estimate of whether the model performs well compared to observations in the dry year, to lend confidence to the further discussions

**Response:** *We evaluated and will include in the results section that the model validations for the dry years (2014, 2015, 2018) even result in better qualities of fit ($r^2$=0.84, RMSE=0.63 t ha$^{-1}$) than for the whole dataset.*

**RC1:** Section 3.4 It would help to have some sort of measure of whether differences are significant

**Response:** *This is the case and already mentioned in the text. All p-values, determined as described in Sec. 2.4, for yield differences between years per number of cuts are smaller than 0.01. However, significance is not a very meaningful measure in the context of large datasets (comp. Sec. 2.4), which is why we did not emphasise this point more.*

**RC1:** Figure captions – please extend the text to be more descriptive of that is the figures

**Response:** *We thank the referee for the demand on more extensive figure captions. We believe that figures are key and will extend the figure legends in a revised version.*

**RC1:** Section 3.5 I do not fully understand the logic behind a corelative analysis of model simulations. Surely since LDNDC is a process-based model, internal drivers can be illustrated based on model output variables and process understanding.

**Response:** *We fully agree that it is an interesting question, if correlative analysis of model inputs and results is necessary and meaningful, since in principle processes are included intrinsically. In complex models like LandscapeDNDC, which incorporate many partly non-linear processes on different scales, it is not possible to follow the matter fluxes continuously, in particular for many simulations. The use of correlations allows to extract, which relations arise in the dataset created through a complex interplay of many processes. These extracted direct relations capture mean behavior and are understandable and applicable within simpler empirical models. For instance, for estimations of yields based on elevation as employed in [Richner et al., 2017], or for look-up tables based on soil quality related to soil organic carbon [Bayerisches Landesamt für Landwirtschaft, 2018].*

**RC1:** Data availability – I believe that the Copernicus policy requires all data and code to be open

**Response:** *The referee is right, that the Copernicus policy requires all data and code to be open. The LandscapeDNDC source code for released versions of the model can be downloaded at the Radar4KIT database (https://doi.org/10.35097/438; Butterbach-Bahl et al., 2021). The model inputs rely on field information from the Integrated Administration and Control System, which is the basis of field locations, boarder, and management class. This data includes privacy information and thus is protected and we can not publish it. Instead, we will upload the model in- and outputs aggregated onto hexagons for individual numbers of cuts to the Radar4KIT repository. The measurements used for validation and regional evaluation will be uploaded to the Bonares repository. The regional cutting dates will soon appear on the EOC Geoservice.*

**Minor comments**

**RC1:** L50 As well as translocation experiments, there are also an increasing number of rainfall exclusion experiments which do not suffer from any of the caveats discussed here

*Response: Thank you for this information. We will read through the according literature and add information on these interesting experiments here.*

**RC1:** L103 this paragraph detailing previous uses of LDNDC belongs more in the introduction than the model description

*Response: Thank you for noting. We will shift and adapt L103-113 to the introduction.*

**RC1:** L117 a 2 year spin up seems extremely short for e.g. soil C and N stocks, what was the reasoning behind this?

*Response: The employed soilchemistry module Metr$^x$ in LandscapeDNDC is set up to adjust soil carbon and nitrogen pool distributions during the first two years of a simulation to align with anticipated annual carbon and nitrogen gains or losses. In this study, we applied the default setting, which assumes equilibrium conditions with no larger net gains or losses. This approach minimizes inital steep gradients of carbon and nitrogen pools, allowing the short explicit spin-up phase.*

**RC1:** L160 what is the spatial resolution for the regional simulation?

*Response: The spatial resolution of our simulations is field-scale, i.e. for every field an individual simulation is run, compare Sec. 2.3. Field areas range from 0.006 ha to 49 ha with a mean value of 1.2 ha. We will include these values in the revised version of the manuscript.*

**RC1:** L161 is there a reference for this data?

*Response: We received this data from the Landesamt für Landwirtschaft in Bavaria. In previous studies (Hänsel et al., 2023), this data was referenced with "IACS., 2014. Integrated Administration and Control System as defined in Commission Implementing Regulation (EU) No 809/2014.", which is the law it is based on. We will add this reference to the manuscript.*

**RC1:** L181 a brief description of how cutting dates were derived would be helpful

*Response: We will add a couple of sentences answering this point: Grassland cutting dates were derived from optical satellite time series data acquired by the Sentinel-2 sensors. Within the approach, which is outlined in more detail in Reinermann et al. 2022, a thresholding rule-set is applied to the Enhanced Vegetation Index derived from the pre-processed Sentinel-2 time series per 10 m x 10 m pixel. The thresholds were calibrated with individual parcels in the Ammer region area for which detailed management information was available. The pixel-based cutting dates were aggregated to field level by using the majority vote.*

**RC1:** L242 why hexagons?

*Response: We chose hexagons as the unit of aggregation, since in a hexagon the distances from the edges to the center are more similar than in a square and the variation of data to be aggregated can be expected to be smaller (for instance, lower climatic gradients per aggregation unit). In other words, a hexagon is more similar to a circle than a square, but still space-filling. For further explanations, see Birch et al., 2007.*

**RC1:** L319 why do MAT and MAP need to be aggregated by field area? Especially the temperature

***Response:*** *This only makes minor differences and is done for consistency, since the aggregated yields are also weighted by field area.*

**RC1:** Figure 6 is this averaged over all fields or averaged over all cuts?

***Response:*** *In Fig. 6, data is averaged over all fields per number of cuts, i.e. all fields with a certain number of cuts are taken together from all fields as a subset and mean values are derived. We will clarify this in the figure caption.*

*Mentioned references*

*Bayerisches Landesamt für Landwirtschaft: Leitfaden für die Düngung von Acker- und Grünland Gelbes Heft., https://www.lfl.bayern.de/publikationen/informationen/040117/index.php, 2018.*

*Butterbach-Bahl, K., Grote, R., Haas, E.: LandscapeDNDC (v1.30.4). Karlsruhe Institute of Technology (KIT). DOI: 10.35097/438, 2021.*

*Birch, C. P.D., Oom, S. P., and Beecham, J. A.: Rectangular and hexagonal grids used for observation, experiment, and simulation in ecology. Ecological Modelling, Vol. 206, No. 3–4., pp. 347–359, 2007.*

*Haensel, M., Scheinpflug, L., Riebl, R., Lohse, E. J., Röder, N, Koellner, T.: Policy instruments and their success in preserving temperate grassland : Evidence from 16 years of implementation. In: Land Use Policy. Bd. 132, 106766, 2023.*

*Isotta, F. A., Frei, C., Weilguni, V., Perčec Tadić, M., Lassègues, P., Rudolf, B., Pavan, V., Cacciamani, C., Antolini, G., Ratto, S.M., Munari, M., Micheletti, S., Bonati, V., Lussana, C., Ronchi, C., Panettieri, E., Marigo, G. and Vertačnik, G., The climate of daily precipitation in the Alps: development and analysis of a high-resolution grid dataset from pan-Alpine rain-gauge data. Int. J. Climatol., 34: 1657-1675, 2014.*

*Korell, L., Andrzejak, M., Berger, S., Durka, W., Haider, S., Hensen, I., Herion, Y., Höfner, J., Kindermann, L., Klotz, S., Knight, T. M., Linstädter, A., Madaj, A.-M., Merbach, I., Michalski, S., Plos, C., Roscher, C., Schädler, M., Welk, E., & Auge, H.. Land use modulates resistance of grasslands against future climate and inter-annual climate variability in a large field experiment. Global Change Biology, 30, e17418, 2024.*

*Richner, W., Sinaj, S., Principles of Agricultural Crop Fertilisation in Switzerland (PRIF). Agrar. Schweiz 8. Spezialpublikation, 276p, 2017.*

---

## Author Comment (AC2)

*The referee comments are upright, and our responses are in italics.*

**General comments**

**RC2:** The title of the study sets a high expectation for drought impact on grassland productivity in the pre-Alpine region of Southern Germany. This was also expressed in the stated main objective, which aims to link the modeled yield to the environmental parameters. Overall, the study seems to be lacking and does not meet the title and objective. Focus was mostly given to the model evaluation, rather than equally explaining the soil, climate, and management factors.

***Response:*** *We thank the referee for this general observation. Our impression is that the messages about the interaction of soil, climate, and management, which are given in Sec. 3.5 and Sec. 3.4, are too hidden between more detailed parts of the study, like method description, model validation, and regional evaluation. We are therefore planning to extend the discussion by better linking soil organic carbon and nitrogen dynamics, as well as plant available water in respect to drought and their influence on grassland yields. We will also deepen the results and discussion on individual numbers of cuts and years, like in Fig. 10 for soil organic carbon and elevation.*

**RC2:** The study also needs to further justify the use of LandscapeDNDC. Are there existing similar and related models that can provide the same outputs?

***Response:*** *As a biogeochemical model, LandscapeDNDC simulates not only grassland yields but also nitrogen, carbon, and water fluxes. In contrast to a crop model, the detailed description of soil processes allows the representation of drought effects on nitrogen turnover and plant nitrogen uptake and hence on productivity. Additionally, nitrous oxide emissions and nitrate leaching are included in the paper (compare Fig. A4 and A5). In a revised version of the manuscript, we will strengthen the discussion of drought effects on nitrogen fluxes. There are other models, like daycent, which can provide similar outputs (compare dos Reis Martins et al., 2024). We still believe that LandscapeDNDC is the best model for this study, since it is calibrated on the lysimeter data within the study region. We will add a couple of sentences regarding the justification of the use of LandscapeDNDC in the revised manuscript.*

**RC2:** Similarly, the study can further give highlights to the Sentinel-2 extracted cutting dates. How is this better than using SAR-based, or SAR-and-MSI extraction methods for cutting dates? The ability to generate the grassland management information for a large-scale study is indeed significant.

***Response:*** *We thank the referee for the question. It was partly answered in Reinermann et al., 2022, where it was found that the additional inclusion of SAR (Sentinel-1) does not improve the detection of cutting events. Please note, that the cutting data was not*

*generated in this study, which is why we do not want to include too many details on the methods.*

**Specific comments**

**RC2:** In relation again to the title, the study could have quantified the drought events with indices such as SPIE or SPI. For instance, the opportunity to provide more information about drought can allow for a better visualization of the impacts in 2018. How does the study define drought? Is it simple related to temperature? Drought is a continuous phenomenon that ignores a defined border of years. Certain drought events may start in year 1, and end in year 2. Instead of annual assessment, would a seasonal assessment provide more realistic results? Such is the argument with the increased spatial resolution. Simply identifying 2018 as a drought year limits the degree of comparison with 2019, and 2020.

*Response: We thank the referee for the suggestion to employ drought indices. We agree that our analysis of the drought severity in 2018 was rather superficial and we are very motivated to improve this in a revised manuscript. We are planning to determine drought indices (SPI, SPIE, or similar) per field and clearly define drought in a more quantitative manner. This framework will be used to extract the periods and areas of drought, which can then be contrasted to the non-drought periods. We will also include the drought indices in the correlation analysis. We hope that this approach will pin-down more clearly, which effects stem from drought and how soil (carbon, nitrogen, water dynamics), management, and climate factors influence yields under drought.*

**RC2:** The study has the potential to show the influence of various factors and the importance of incorporating grassland management when determining drought impact. It should maximize the available data and add information. For example, the time of cuts can also be determined. The harvest of biomass in grasslands is related to the optimum growth of the vegetation, the time of harvest may reflect adaptive practices (management) by farmers.

*Response *: The referee suggests to relate the time of cuts with drought and yields. We looked into this already and tried to elaborate if the trends in yield increase with day of cut differ between the years, which is shown in Fig. 1 and 2 for 3 and 5 cuts. We did not find convincing trends. The only clear feature was an increased number of cuts in 2018 compared to the other years (Sec. 2.3.2). We will pick up on these analyses after determining the drought indices, since their inclusion might lead to more conclusive results. For instance, by only contrasting fields under drought conditions to the ones not suffering from drought.*

[Figure]

Figure 1: For all field-year pairs with 3 cuts in the year, the harvested dry weight biomass of individual cutting events is plotted against the day of the year of the cut. The years 2018, 2019, and 2020 are shown in different colors. On the right hand side of the plot, the Pearson correlation coefficients, coefficients of determination, and the slope of the regression are given.

[Figure]

Figure 2: Same as Fig. 1 but for all field-year pairs with 5 cutting events.

**RC2:** The environmental factors and results can be summarized in table forms. These can show what were all the considered factors, sources, and resolution.

*Response: We will add such a summarizing table.*

**Technical corrections and minor comments**

**RC2:** These are some observed writing concerns.

**RC2:** Line 37 missing sentence or phrase. Or the need to remove of parenthesis for in-text citations that are part of the sentence.

*Response: We will reformulate the sentence to: "As additional ecosystem services, especially extensively used grasslands support biodiversity (Wilson et al., 2012; Väre et al., 2003) and permanent grasslands support water retention and reduce erosion (White, 2000; Bengtsson et al., 2019)." to clarify the references.*

**RC2:** Line 54 Missing year of cited study by De Boeck et al.

*Response: The year 2016 will be added.*

**RC2:** Line 55 Consistency with the use of space between values and units of measurement. Some lack space, others have space.

*Response:* *We will add spaces.*

**RC2:** Line 93 For the n=28202; how high is the spatial resolution as compared to other European scale studies?

*Response:* *As a comparison, in Carozzi et al., 2022 European grassland simulations were performed on a 0.25° grid relating to squared cells of 772 km². The Ammer catchment has a total area of approximately 4600 km² for which we performed 28202 simulations on field-scale ranging up to 0.5 km² as a maximum with a mean simulation domain size of 0.02 km² (2 ha). We thank the referee for the question and will include the comparison in a revised version of the manuscript.*

**RC2:** Line 175 The table tile on top of the table.

*Response:* *All table captions will be shifted to the top of the table.*

**RC2:** Line 238 missing word or phrase

*Response:* *The sentence will be corrected to: "These values are multiplied by the amount of rainfall at days of precipitation and nitrogen loads are added to the first soil layer."*

**RC2:** Line 330 Missing figure number

*Response:* *Figure number 4 will be added.*

**RC2:** Line 346 In-text citation before the presentation of the figure

*Response:* *The sentence will be restructured to: "In Fig. 5, the monthly mean temperatures (a) and sums of precipitation (b) averaged over all fields are shown."*

**RC2:** Consistency, for some parts the corresponding letters were written before the data (Line 317); while for others these were written after (Line 343)

*Response:* *Thank you for noticing. We will always mention the letters after the data in the revised manuscript.*

**RC2:** Line 320 Why were hexagons used? What are the unit of the other input values? For instance, Sentinel 2 pixels are in squares.

*Response:* *We chose hexagons as the unit of aggregation, since in a hexagon the distances from the edges to the center are more similar than in a square and the variation of data to be aggregated can be expected to be smaller (for instance, lower climatic gradients per aggregation unit). In other words, a hexagon is more similar to a circle than a square, but still space-filling. For further explanations, see Birch et al., 2007. Model inputs were given as follows: climate data 500x500 m grid of virtual stations, field scale*

*cutting dates (aggregated beforehand from 10 m resolution, Reinermann et al. 2022 and 2023), soil data was provided at varying polygon sizes (from sub-field to larger-field).*

**RC2:** Line 374 The decreasing trend in mean yield in 2018 may also be related to the optimal vegetation growth. The timing of cuts might be relevant.

***Response:*** *see answer marked with \* above*

**RC2:** Line 436 It is believed that the basic statistical assumptions were tested. Maybe results can be provided as supplementary material or appendix.

***Response:*** *This is indeed true. The according p-values will be provided in the Appendix in a revised version.*

**RC2:** Line 440 For the multiple correlation, why was Principal Component Analysis (PCA) or its equivalent not utilized? It is better to show how all parameters were related.

***Response:*** *We thank the referee for the suggestion of principle component analysis (PCA) as the tool for multiple correlation analysis. We believe that PCA is not beneficial here, because, on the one hand, our data set contains outliers (compare Fig. 10c) and partly discrete values (compare Fig. 10 a and b), which challenge the PCA, on the other hand, it is not straightforward to interpret the results and employ them in empirical models.*

**RC2:** A number of results were not included in the paper, maybe these can be included as supplementary materials or in the appendix.

***Response:*** *We will include results for which no figures were given in the appendix of the revised version. A few of these are shown already below in Fig. 3 – Fig. 5. Additionally, we will include the regressions of all field-year pairs with 3 and 5 cuts against soil organic carbon, mean annual temperature, elevation, as well as a plot linking soil organic carbon and nitrogen content.*

[Figure]

Figure 3: Correlation matrices for field data linking soil properties (BD, SOC, pH, plant available water [wcdif]), elevation, climatic parameters (MAT, MAP) averaged over 2018–2020, as well as the number of cuts in the year (#cuts) with yearly harvested biomass (yields). Coefficients written (not) in bold indicate a p-value smaller (equal or larger) 0.01. All fields in 2018 (a), 2019 (b), and 2020 (c) are included.

[Figure]

Figure 4: Yields as DWBM in individual years plotted against SOC (a) and elevation (b). Every point represents a single field-year pair. Fitted regression and associated correlation coefficients and slopes are given in the boxes for each year. All regressions are significant (p<0.0005).

[Figure]

*Figure 5: Annual yields of all field-year pairs are plotted against the number of cuts. Correlation coefficient, coefficient of determination, and slope are given in the box.*

**RC2:** Line 478 I commend the possible inclusion of plant functional traits.

***Response:*** *We are not sure about this comment, since line 478 deals with plant functional types and not traits. Regarding plant functional traits, these are covered by model species input parameters and therefore included in the simulations. If the referee suggests plant functional types, we want to point out, that there is only sparse data on plant species composition linked to other properties like management or elevation, and yields. However, this kind of data would be required to come up with regional model inputs and model validation without adding further uncertainties. We think, that future developments in remote sensing models on plant species composition may allow a regional differentiation between plant functional types. These aspects were already included in the discussion section.*

**RC2:** Line 580 Review the sentence, "Another reasons" – very minor

***Response:*** *It should be: "Another reason for rather low yields in 2019 is the spatially uneven distribution of precipitation (compare Fig. 4)."*

**RC2:** Line 620 Use of "a" and "an" – very minor

***Response:*** *Will be adapted.*

*Mentioned references*

Birch, C. P.D., Oom, S. P., and Beecham, J. A.: Rectangular and hexagonal grids used for observation, experiment, and simulation in ecology. Ecological Modelling, Vol. 206, No. 3–4., pp. 347–359, 2007.

Carozzi, M., Martin, R., Klumpp, K., and Massad, R. S.: Effects of climate change in European croplands and grasslands: productivity, greenhouse gas balance and soil carbon storage, Biogeosciences, 19, 3021–3050, https://doi.org/10.5194/bg-19-3021-2022, 2022.

dos Reis Martins, M., Ammann, C., Boos, C., Clanca, P., Kiese R., Wolf, B., Keel, S.G.: Reducing N fertilization in the framework of the European Farm to Fork strategy under global change: Impacts on yields, N2O emissions and N leaching of temperate grasslands in the Alpine region. Agricultural Systems, Vol. 219, 104036, 2024.

Reinermann, S., Gessner, U., Asam, S., Ullmann, T., Schucknecht, A., and Kuenzer, C.: Detection of Grassland Mowing Events for Germany by Combining Sentinel-1 and Sentinel-2 Time Series, Remote Sensing, 14, 1647, https://doi.org/10.3390/rs14071647, 2022.

Reinermann, S., Asam, S., Gessner, U., Ullmann, T., and Kuenzer, C.: Multi-annual grassland mowing dynamics in Germany: spatio-temporal patterns and the influence of climate, topographic and socio-political conditions, Front. Environ. Sci., 11, https://doi.org/10.3389/fenvs.2023.1040551, 2023.